# DBGL: Decay-aware Bipartite Graph Learning for Irregular Medical Time Series

## Abstract

Irregular Medical Time Series (IMTS) are of great importance in the healthcare domain to better understand the patient's condition. However, the inherent temporal irregularity, arising from heterogeneous sampling rates, asynchronous observations, and variable gaps, poses significant challenges for reliable modeling. Existing methods distort the **temporal sampling irregularity** and missing pattern, while failing to capture **variable decay irregularity** in the clinical domain, leading to suboptimal representation. To address these limitations, we introduce DBGL: Decay-Aware Bipartite Graph Learning for Irregular Medical Time Series. DBGL first introduces a patient–variable bipartite graph that simultaneously captures irregular sampling patterns without artificial alignment and adaptively models variable relationships for temporal sampling irregularity modeling, enhancing representation learning. To model variable decay irregularity, DBGL designs a novel node-specific temporal decay encoding mechanism that enables each variable to decay at different rates based on sampling interval, yielding a more accurate and faithful representation of irregular temporal dynamics. We evaluate the performance of DBGL on four publicly available datasets: P19, Physionet, MIMIC-III, and P12. Results show that DBGL outperforms all baselines, and our code is also available in the supplementary material.

## 1 Introduction

Medical time series (MTS) from electronic health records (EHRs) are central to healthcare analysis but are often irregular due to clinical workflows, costs, and patient conditions (Sun et al. (2020); Chen et al. (2025a)). This irregularity arises from heterogeneous sampling frequencies (e.g., continuous heart rate vs. hourly blood tests), asynchronous observations, and variable temporal intervals, resulting in irregular medical time series (IMTS) (Shukla & Marlin (2018)). While it poses challenges for conventional modeling (Wu et al. (2022)), it also contains informative signals: clinicians alter measurement frequency and ordering in response to concern or changing severity, and the resulting patterns of sampling and missingness (e.g., bursts around events, long gaps during stability) can be exploited as prognostic and diagnostic cues. Successfully addressing these challenges is of great significance for early risk prediction, patient state tracking, and reliable clinical decision support (Tan et al. (2020a)).

Modeling irregular multivariate time series (IMTS) is challenging not only because observations occur at heterogeneous rates, but also because these irregularities distort temporal dependencies. This temporal sampling irregularity (Nielsen (1994); Liu et al. (2022)), i.e., uneven and asynchronous measurements, complicates the alignment of variables in a coherent representation space. In addition, different clinical variables evolve at different rates over time, a phenomenon we term variable decay irregularity (Lipton et al. (2016); Che et al. (2018)). Formally, we quantify this effect using a decay rate $\lambda$ estimated from empirical autocorrelations of each variable: higher $\lambda$ indicates fast-changing variables (e.g., heart rate, blood pressure), while lower $\lambda$ corresponds to slower, more stable variables (e.g., creatinine, hemoglobin). This provides a unified, quantitative measure of clinically meaningful differences in temporal dynamics. Representative decay rates estimated on the P12 dataset are shown in Table 1 and more details can be found in the Appendix A.

Irregular sampling is a central challenge in multivariate clinical time-series. Traditional interpolation or resampling pipelines produce regularly spaced sequences but may obscure informative missingness patterns (Chen et al. (2018)). Recent RNN-based (Che et al. (2018); Rajkomar et al. (2018)), diffusion-based (Rubanova et al. (2019); Kidger et al. (2020)), and transformer-based approaches (Zheng et al. (2024); Ren et al. (2024)) incorporate continuous-time embeddings or learned interpolation to mitigate these issues, yet they still operate primarily on flattened sequences and therefore struggle to represent which variable is observed at which time, a key structural signal in clinical data. Bipartite variable–time graphs (Yalavarthi et al. (2024); Liu et al. (2025)) make this structure more explicit by linking variable nodes to time nodes, but they are typically constructed within local windows and do not capture patient-specific sampling behaviors. Moreover, clinical variables naturally exhibit heterogeneous temporal dynamics (Tan et al. (2020b)), some change within minutes, others over hours or days, yet most existing models encode time uniformly and lack explicit mechanisms for variable-specific decay.

Table 1: Representative variable decay rates ($\lambda$) on the P12 dataset.

| Variable | $\lambda$ | Variable | $\lambda$ |
|---|---|---|---|
| HR | 1.73 | Temp | 0.0265 |
| Creatinine | 0.1399 | Lactate | 13.00 |
| ALP | 14.71 | PaCO2 | 0.0225 |
| ALT | 14.62 | AST | 13.89 |

In summary, two main limitations remain in existing approaches. First, preprocessing or flattened sequence modeling can obscure inherent irregularity: resampling may eliminate informative missingness, and sequence-based variable modeling captures limited asynchronous interactions between patients and variables. Second, temporal dynamics are often treated uniformly across variables, ignoring that clinical variables evolve at heterogeneous rates and exhibit variable-specific sensitivity to elapsed time. This simplification can lead to incomplete or less informative patient representations for downstream clinical tasks.

To address these limitations, we propose **D**ecay-aware **B**ipartite **G**raph **L**earning (**DBGL**) with two key components. First, for temporal sampling irregularity, DBGL models IMTS as a sequence of patient–variable bipartite graphs, where at each time step, edges are created only for observed variables. This design preserves the true observation structure and explicitly encodes irregular sampling patterns without artificial alignment. Second, we introduce a node-specific temporal decay encoding mechanism, which decays patient hidden states according to the elapsed time and then updates them with new variable observations. This enables each variable to "forget" at clinically realistic speeds while maintaining a continuously evolving patient state. We evaluate DBGL on four publicly available clinical datasets. Results demonstrate that DBGL consistently outperforms all baselines across tasks. Detailed analyses and ablation studies further reveal the key factors driving its performance gains. Our main contributions can be summarized as:

- We propose DBGL, a novel framework that embeds irregular sampling patterns directly into the graph topology. By constructing each time step as a patient–variable bipartite graph, DBGL preserves informative observation dependencies without artificial alignment. Through graph message passing, correlations among variables are adaptively aggregated into patient nodes, yielding more expressive and patient-specific representations.

- We introduce a novel temporal decay encoding mechanism with node-specific updates to model variable-dependent decay irregularities in IMTS. Unlike a uniform temporal discount, our design allows each variable to follow its own adaptive decay trajectory. At each time step, the hidden state decays according to the sampling interval and is then updated based on new observations. This design captures fine-grained and heterogeneous temporal dynamics across variables for a continuously evolving representation of patient states.

- We conduct comprehensive experiments on four public clinical datasets. Experimental results demonstrate that DBGL achieves superior performance compared to existing methods on all datasets, showcasing its robustness and adaptability across various scenarios.

## 2 RELATED WORK

### 2.1 IRREGULAR TIME SERIES ANALYSIS

Irregularly sampled time series are common in healthcare, finance, and transportation. Existing approaches mainly fall into two categories. Interpolation-based methods, such as kernel smoothing, Gaussian processes (Tan et al. (2021)), or temporal aggregation (Ma et al. (2020)), resample

irregular observations onto regular grids, but often distort original sampling patterns and obscure informative missingness. Direct modeling approaches avoid resampling by explicitly incorporating time intervals, including recurrent networks for non-uniform gaps (Che et al. (2018)), temporal embeddings for arbitrary timestamps (Horn et al. (2020); Shukla & Marlin (2021a)), and neural ODEs for continuous-time dynamics (Schirmer et al. (2022); Chen et al. (2023)). Attention- and graph-based architectures have also been explored for capturing long-range dependencies and inter-variable relations. Yet, most prior work focuses on local irregularities, leaving asynchronous variable dependencies and variable-specific decay largely unmodeled.

## 2.2 GRAPH-BASED METHODS FOR CLINICAL TIME SERIES

Graph neural networks (GNNs) naturally capture dependencies among variables, patients, and temporal contexts in clinical time series. Prior studies have applied GNNs to model inter-variable relations for prediction (Escudero-Arnanz et al. (2024)), capture spatial-temporal EEG dependencies (Varatharajah et al. (2017); Tang et al. (2021)), reflect dependencies among clinical variables (Zheng et al. (2023)), predict patient interventions (Xu et al. (2024)), model dynamic multi-resolution temporal-spatial dependencies (Fan et al. (2025)), and integrate textual medical knowledge (Luo et al. (2024)). Despite these advances, such approaches do not explicitly address irregular medical time series, lacking mechanisms to leverage the inherent sampling and decay irregularities in IMTS. To bridge this gap, we introduce **DBGL**, a decay-aware bipartite graph learning framework that explicitly encodes temporal decay and sampling irregularities, enabling more robust and faithful representation learning for irregular clinical time series.

## 3 METHODOLOGY

### 3.1 OVERVIEW

DBGL tackles the sampling irregularity of IMTS by representing each time step as a patient–variable bipartite graph, where irregular sampling patterns are directly encoded into the graph structure. This formulation preserves informative dependencies among observations without requiring artificial temporal alignment, which is crucial for modeling non-uniformly sampled data. To further capture variable-specific temporal decay, DBGL employs a novel temporal decay encoding mechanism with node-specific update that adaptively modulates each variable's influence over time, enabling the model to learn fine-grained temporal dynamics and coherent patient trajectories. Moreover, a learnable codebook is incorporated to represent common latent patient states, facilitating more expressive and generalizable representations. The overall framework of DBGL is illustrated in Figure 1.

### 3.2 PATIENT-VARIABLE BIPARTITE GRAPH

Temporal sampling irregularity is a fundamental challenge in medical time series, as different variables are measured at heterogeneous and often unpredictable intervals. Existing approaches typically resort to resampling or imputation, which may distort temporal dynamics and obscure informative structures. Recent graph-based methods attempt to address this issue by constructing fully connected variable graphs and masking adjacency entries according to observed variables. However, such masking mainly encodes observation availability and fails to effectively capture the sampling irregular patterns and model the interaction between patients and variables explicitly.

**Bipartite Patient-Variable Graph Construction.** To overcome these limitations, we represent medical time series as a sequence of patient-variable bipartite graphs. Specifically, for each time step, DBGL transforms the medical time series features $\{X^p\}_{p=1}^{N}$ and observation flag matrix $M$ to an undirected bipartite patient-variable graph $G_t = (V, E)$. Here, the node set $V = V_p \cup V_v$, consisting of $p$ patient nodes $V_p = \{u_1, u_2, ..., u_p\}$ and $n$ variable nodes $V_v = \{v_1, v_2, ..., v_n\}$. The edge set $E$ consists of the current observed state, which is determined using the observation flag matrix $M$, and is referred to as the adjacency matrix. Specifically, an edge $e_{u_p,v_n}$ will be constructed if the patient $p$ has the data for the variable $n$. In this way, the whole IMTS will be modelled as a series ographsph $G = \{G_1, G_2, ..., G_T\}$, where $T$ is the total time step.

**Embedding Initialization.** For each observed entry, we obtain an edge embedding by summing three components: a value embedding derived from the measurement, a time embedding encoding

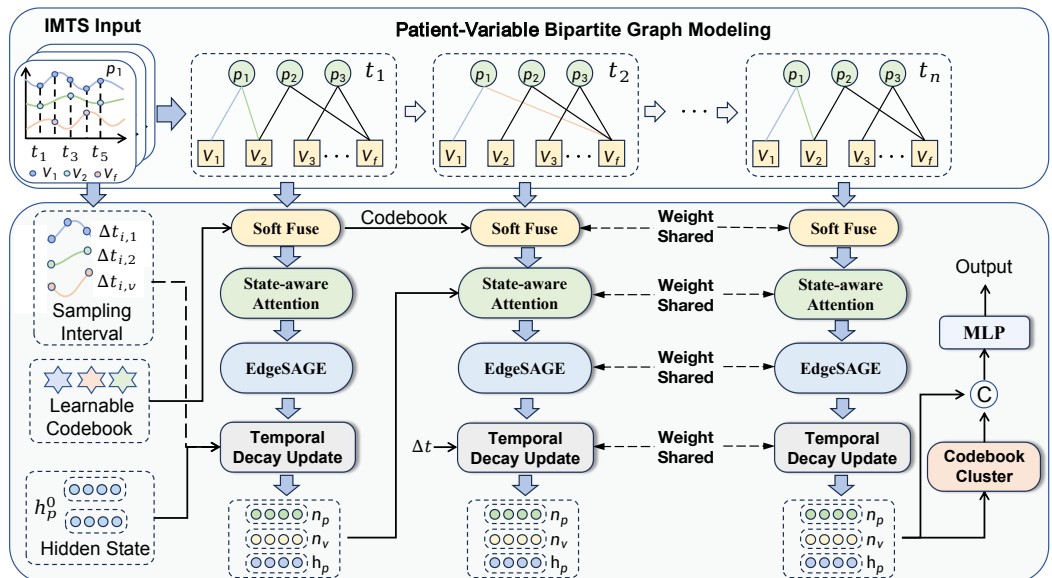

Figure 1: Framework of our proposed DBGL. DBGL first models the IMTS as patient-variable bipartite graphs. Furthermore, a novel mechanism called temporal decay encoding with node-specific updates is used to obtain a robust representation for classification.

absolute sampling intervals via linear and sinusoidal projections (Kazemi et al. (2019)), and a learnable variable type embedding. Unobserved entries are masked out according to $M$. In addition, patient nodes are initialized as constant vectors during each forward pass, while variable nodes are learnable parameters initialized with Xavier uniform distribution. Together, these node and edge embeddings provide the bipartite graph with both observation-specific semantics and adaptable representations of patients and variables.

**Message Passing over the Bipartite Graph.** Based on the constructed patient-variable bipartite graph, we employ a multi-layer EdgeSAGE network to propagate and aggregate information across patients, variables, and their temporal observations. At the $l$-th layer, messages are first constructed by jointly encoding neighbor node states and edge attributes as Eq. 1

$$m_{i,j}^{(l)} = \sigma\left(W_m^{(l)} \cdot [v_j^{(l)} || e_{ij}^{(l)}] + b_m\right), \quad \widetilde{v}_i^{(l)} = \sum_{j \in N_i} m_{i,j}^{(l)}, \tag{1}$$

where $||$ denotes concatenation, $v_j^{(l)}$ is the embedding of neighbor node $j$, $e_{ij}^{(l)}$ is the edge embedding of edge linking node $i$ and $j$, $\sigma$ is the ReLU activation, and $N_i$ is the neighbor node set of node $i$. The final output of this module is the updated node and edge states, $v_{p,n}^t$ and $e_{p,n}^t$, as shown in Eq. 2:

$$v_i^{(l+1)} = \sigma(W_h^{(l)} \cdot [v_i^{(l)} || m_{i,j}^{(l)}] + b), \quad e_{i,j}^{(l+1)} = e_{i,j}^{(l)} + \sigma(W_h^{(l)} \cdot [v_i^{(l+1)} || v_j^{(l+1)} || e_{i,j}^{(l)}] + b). \tag{2}$$

Unlike conventional temporal models that either resample or impute irregularly observed data, our message passing explicitly conditions edge embeddings on the actual observation times. This allows patient nodes to continuously integrate variable-specific dynamics at irregular intervals, while edge updates preserve temporal dependencies between observations. As a result, DBGL effectively mitigates the temporal sampling irregularity of IMTS by adaptively aligning patient trajectories with the true clinical observation patterns.

### 3.3 Temporal Decay Encoding Mechanism

**Temporal Decay Encoding.** To address the variable-specific temporal irregularity in IMTS, we design a temporal decay encoding mechanism that adaptively adjusts each variable's hidden state according to its sampling interval. We first initialize the state representation of each variable for every patient, $h^{(0)}$, as zero, and then design a temporal decay encoding mechanism to update these

states at each time step based on the representations aggregated from the bipartite graph and the previous hidden state.

Specifically, at each time step $t$, given $h_{p,n}^{(t-1)}$, the previous hidden state of the $n$-th variable of the patient $p$, we compute a decay rate $\lambda_{p,n}^t$ based on the current edge representations $e_{p,n}^t$ through a small MLP, and introduce a sampling decay factor is then determined by the elapsed interval $\Delta t$ for hidden state decay factor $\gamma_{p,n}^t$. This process is illustrated as Eq. 3:

$$\lambda_{p,n}^t = \text{Softplus}\left(\text{MLP}(e_{p,n}^t)\right), \quad \gamma_{p,n}^t = \exp(-\lambda_{p,n}^t \cdot \Delta t), \tag{3}$$

and then the decayed hidden state can be obtained by $\hat{h}_{p,n}^{t-1} = \gamma_{p,n}^t * h_{p,n}^{t-1}$. This factor ensures that the hidden state decays smoothly with longer gaps, naturally reflecting the uncertainty caused by irregular sampling. The decayed state is then combined with the new observation through a gated update mechanism with the sigmoid function $\sigma$, as shown in Eq. 4:

$$r_{p,n}^t = \sigma\left(W_r[e_{p,n}^t || \hat{h}_{p,n}^{t-1}\right), \quad h_{p,n}^t = (1 - r_{p,n}^t) \cdot \hat{h}_{p,n}^{t-1} + r_{p,n}^t \cdot e_{p,n}^t. \tag{4}$$

Notably, the elapsed sampling interval $\Delta t$ is obtained by the previous or next observation timestamp of variable $v$, which is shown as Eq. 5:

$$\Delta t_{i,v} = \begin{cases} \dfrac{(t_{i,v} - t_{i-1,v}) + (t_{i+1,v} - t_{i,v})}{2}, & \text{if both } t_{i-1,v} \text{ and } t_{i+1,v} \text{ exist,} \\[2mm] t_{i,v} - t_{i-1,v}, & \text{if } t_{i+1,v} \text{ does not exist,} \\[2mm] t_{i+1,v} - t_{i,v}, & \text{if } t_{i-1,v} \text{ does not exist,} \\[2mm] \dfrac{t_{\max}}{2}, & \text{if neither } t_{i-1,v} \text{ nor } t_{i+1,v} \text{ exists.} \end{cases} \tag{5}$$

**Node-specific Updates.** To further leverage the historical states of the variables of a patient, we design a hidden state-aware node-specific attention mechanism that performs node-specific updates before patient nodes enter the graph network. Since there is no hidden state in the first time step, this mechanism is used from the second time step. Specifically, given the previous hidden state of all variables $\mathbf{h}^{t-1}$, our state-aware attention output is computed as Eq. 6:

$$\mathbf{a}_p^t = \text{softmax}\left(\frac{\mathbf{v}_p^t (\mathbf{h}^{t-1})^\top}{\sqrt{d}}\right)\mathbf{h}^{t-1}, \quad \mathbf{v}_p^t = W_{\text{proj}} \cdot \mathbf{a}_p^t, \tag{6}$$

where the softmax is applied over variable nodes to capture variable-specific temporal context, and the attended feature is then projected to update the patient node embedding. This node-specific attention allows each patient node to selectively incorporate the most relevant information from its historical variable states. By doing so, the patient representation is enriched with fine-grained, variable-wise temporal context, which is particularly important for handling irregularly sampled medical time series and ensuring that subsequent graph message passing operates on a temporally informed patient embedding.

### 3.4 COMMON STATE CODEBOOK LEARNING.

Although the bipartite graph construction and temporal decay encoding mechanism effectively capture irregular sampling patterns of IMTS, the learned representations of patient and variable nodes may still be limited in learning the similarities or common patterns among different patients. To address this, we introduce a learnable soft-codebook that serves as a set of global prototypes to regularize and align node embeddings.

Formally, let $\mathbf{C} \in \mathbb{R}^{K \times d}$ denote the codebook with $K$ learnable prototypes of dimension $d$. Given the node embeddings $\mathbf{g}_i = [v_p || v_n] \in \mathbb{R}^d$, we first compute the cosine similarity between each node and all codebook entries. Then the quantized representation can be obtained as a weighted sum over prototypes. This process is illustrated as Eq. 7:

$$s_{i,k} = \frac{\mathbf{g}_i}{\|\mathbf{g}_i\|} \cdot \frac{\mathbf{c}_k}{\|\mathbf{c}_k\|}, \quad \mathbf{c}_k \in \mathbf{C}, \quad \mathbf{g}_i^{\text{quant}} = \sum_{k=1}^{K} w_{i,k} \cdot \mathbf{c}_k. \tag{7}$$

Finally, we apply a residual update with an adaptive scaling factor to balance the original embedding and the quantized representation as Eq. 8

$$\mathbf{g}_i \leftarrow \mathbf{g}_i + \alpha_i \cdot \mathbf{g}_i^{\text{quant}}, \quad \alpha_i = \frac{\|\mathbf{g}_i^{\text{quant}}\|}{\|\mathbf{g}_i\| + \epsilon}. \tag{8}$$

This learnable codebook plays two key roles: (1) it compresses noisy continuous representations into a compact set of prototypes, improving robustness to irregular sampling; and (2) it aligns variable dynamics across patients by projecting them onto shared prototypes, thereby enhancing the generalizability of learned patient representations.

## 3.5 TRAINING AND INFERENCE

After temporal updating and graph propagation, DBGL produces the patient node embeddings $\mathbf{v} \in \mathbb{R}^{B \times d}$ and the per-batch, per-variable hidden states $\mathbf{h} \in \mathbb{R}^{B \times N \times d}$, where $B$ is the batch size, $N$ is the variable count, and $d$ is the hidden dimension. To emphasize the contribution of observed variables, we first apply a re-weighting operation based on the observation mask, followed by flattening the variable states into a single vector, as shown in Eq. 9:

$$\mathbf{h} \leftarrow \mathbf{h} + \mathbf{h} \cdot \text{softmax}\big(\sum_v \text{Mask}_v\big), \quad \mathbf{h} \leftarrow \text{reshape}(\mathbf{h}, [B, -1]), \tag{9}$$

where $\text{Mask}_v$ indicates the observation status of variable $v$. This re-weighting ensures that frequently observed variables have a stronger influence on the patient representation, while the flattening step facilitates integration with subsequent codebook retrieval and classification.

Additionally, to better enable the codebook itself to capture underlying patient state patterns, we compute the similarity between patient node embeddings $\mathbf{g}_p$ and the learnable codebook $\mathbf{C}$ via cosine similarity and select the most similar codebook entry for each patient node, as Eq. 10:

$$\text{Sim} = \frac{\mathbf{g}_p}{\|\mathbf{g}_p\|} \cdot \frac{\mathbf{C}^\top}{\|\mathbf{C}\|}, \quad \mathbf{c}_p = \arg\max_j \text{Sim}_{p,j}, \tag{10}$$

where $\mathbf{c}_p$ denotes the retrieved code vector for patient $p$. The learnable codebook not only provides a mechanism to constrain patient embeddings via retrieval but also evolves into a set of representative latent states during training.

Finally, we concatenate the patient node embedding, its matched code vector, and the flattened hidden representation to form the final representation for classification:

$$\mathbf{z}_p = [\mathbf{g}_p; \mathbf{c}_p; \mathbf{h}], \quad \hat{\mathbf{y}}_p = MLP(\mathbf{z}_p) \tag{11}$$

which is then fed into a classification layer to predict the clinical outcome $\hat{\mathbf{y}}_p$, and Cross Entropy loss is used for optimization.

# 4 EXPERIMENT

## 4.1 EXPERIMENTAL SETUP

**Datasets and baselines.** We conduct extensive experiments on four widely used irregular medical time series (IMTS) datasets: P19 (Reyna et al. (2020)), PhysioNet (Goldberger et al. (2000)), MIMIC-III (Johnson et al. (2016)), and P12 (Silva et al. (2012)). Following prior work (Shukla & Marlin (2021a)), PhysioNet is considered a reduced version of P12. To comprehensively evaluate the effectiveness of DBGL, we compare DBGL against two categories of baselines: non-graphical methods and graph-based methods. Non-graphical methods include GRU-D (Che et al. (2018)), ODE-RNN (Rubanova et al. (2019)), IP-Net (Shukla & Marlin (2019)), SeFT (Horn et al. (2020)), mTAND (Shukla & Marlin (2021b)), DGM$^2$-O (Wu et al. (2021)), StraTS (Tipirneni & Reddy (2022)), DuETT (Labach et al. (2023)), ViTST (Li et al. (2023)) and Warpformer (Zhang et al. (2023)). Graph-based methods include MTGNN (Wu et al. (2020)), Raindrop (Zhang et al. (2021)), ISIM (Chen et al. (2025b)), TimeCHEAT (Liu et al. (2025)), and KEDGN (Luo et al. (2024)). Details about these datasets and baselines can be found in Appendix C and Appendix E.

**Implementation.** We train DBGL using the Adam optimizer (Kingma (2014)) for 30 epochs, employing an early stopping strategy with a patience of 5 epochs to prevent overfitting. To better verify

Table 2: Performance comparison (AUROC & AUPRC, %) on four clinical datasets. The best results are highlighted in **bold** and the second-best results are in underlined.

| Methods | | P19 | | Physionet | | MIMIC-III | | P12 | |
|---|---|---|---|---|---|---|---|---|---|
| | | AUROC | AUPRC | AUROC | AUPRC | AUROC | AUPRC | AUROC | AUPRC |
| Non-Graph | GRU-D | $88.7 \pm 1.2$ | $57.6 \pm 2.3$ | $79.1 \pm 6.9$ | $42.7 \pm 7.2$ | $82.2 \pm 1.8$ | $43.3 \pm 2.1$ | $79.6 \pm 0.6$ | $41.7 \pm 1.8$ |
| | ODE-RNN | $87.1 \pm 1.0$ | $52.6 \pm 3.2$ | $75.5 \pm 2.8$ | $33.7 \pm 4.1$ | $81.0 \pm 0.6$ | $42.3 \pm 0.7$ | $78.8 \pm 0.6$ | $37.4 \pm 2.6$ |
| | IP-Net | $90.2 \pm 0.2$ | $58.6 \pm 0.8$ | $86.8 \pm 0.6$ | $55.8 \pm 1.4$ | $84.1 \pm 0.1$ | $47.1 \pm 0.9$ | $83.7 \pm 0.3$ | $46.3 \pm 1.3$ |
| | SeFT | $84.0 \pm 0.3$ | $49.3 \pm 0.5$ | $75.5 \pm 0.2$ | $29.4 \pm 0.9$ | $67.9 \pm 0.2$ | $23.2 \pm 0.4$ | $78.1 \pm 0.5$ | $35.9 \pm 0.8$ |
| | mTAND | $82.9 \pm 0.9$ | $32.3 \pm 1.5$ | $86.9 \pm 1.3$ | $52.5 \pm 1.3$ | $83.8 \pm 0.3$ | $46.6 \pm 0.5$ | $85.3 \pm 0.3$ | $49.3 \pm 1.0$ |
| | $DGM^2$-O | $91.6 \pm 0.5$ | $60.0 \pm 1.3$ | $85.8 \pm 0.7$ | $50.4 \pm 3.2$ | $81.0 \pm 0.9$ | $37.6 \pm 1.1$ | $85.8 \pm 0.1$ | $48.3 \pm 0.7$ |
| | StraTS | $91.2 \pm 0.3$ | $58.4 \pm 1.4$ | $84.9 \pm 1.5$ | $47.3 \pm 5.3$ | $84.4 \pm 0.4$ | $46.4 \pm 0.8$ | $86.7 \pm 0.7$ | $52.1 \pm 1.5$ |
| | DuETT | $88.2 \pm 0.5$ | $56.0 \pm 3.9$ | $81.3 \pm 1.4$ | $44.9 \pm 1.4$ | $78.8 \pm 0.8$ | $34.3 \pm 1.0$ | $83.4 \pm 1.2$ | $45.4 \pm 1.5$ |
| | ViTST | $91.7 \pm 0.1$ | $57.5 \pm 0.7$ | $81.3 \pm 1.9$ | $37.4 \pm 2.9$ | $81.8 \pm 0.3$ | $39.6 \pm 1.3$ | $86.3 \pm 0.1$ | $50.8 \pm 1.5$ |
| | Warpformer | $91.8 \pm 0.4$ | $60.6 \pm 2.6$ | $83.3 \pm 0.7$ | $43.5 \pm 2.3$ | $84.6 \pm 0.5$ | $47.4 \pm 0.9$ | $85.4 \pm 0.5$ | $50.4 \pm 1.5$ |
| Graph-based | MTGNN | $88.5 \pm 1.0$ | $55.8 \pm 1.5$ | $77.1 \pm 4.4$ | $35.4 \pm 7.3$ | $78.5 \pm 2.3$ | $35.2 \pm 3.1$ | $82.1 \pm 1.5$ | $41.8 \pm 2.1$ |
| | Raindrop | $89.4 \pm 0.6$ | $61.2 \pm 1.1$ | $82.7 \pm 1.4$ | $41.2 \pm 3.6$ | $79.8 \pm 1.3$ | $35.2 \pm 1.1$ | $82.2 \pm 1.1$ | $43.3 \pm 2.1$ |
| | ISIM | $91.6 \pm 0.9$ | $59.6 \pm 1.3$ | - | - | - | - | $86.0 \pm 0.3$ | $50.4 \pm 2.1$ |
| | TimeCHEAT | $89.5 \pm 1.9$ | $56.1 \pm 4.6$ | - | - | - | - | $84.5 \pm 0.7$ | $48.2 \pm 1.9$ |
| | KEDGN-Name | $92.3 \pm 1.0$ | $62.5 \pm 0.7$ | $87.9 \pm 1.4$ | $56.0 \pm 3.2$ | $84.8 \pm 0.3$ | $48.4 \pm 1.5$ | $87.1 \pm 0.8$ | $54.1 \pm 2.6$ |
| | KEDGN-Wiki | $92.2 \pm 0.6$ | $62.3 \pm 1.4$ | $88.2 \pm 1.1$ | $57.5 \pm 2.5$ | $84.3 \pm 0.9$ | $47.7 \pm 1.8$ | $87.0 \pm 0.3$ | $53.1 \pm 0.5$ |
| | KEDGN-ChatGPT | $92.2 \pm 0.5$ | $62.0 \pm 1.3$ | $87.9 \pm 0.2$ | $57.1 \pm 1.8$ | $85.1 \pm 0.3$ | $48.3 \pm 1.6$ | $87.8 \pm 0.5$ | $54.5 \pm 1.5$ |
| Ours | DBGL | **$93.3 \pm 0.5$** | **$66.3 \pm 1.4$** | **$89.1 \pm 0.3$** | **$60.8 \pm 2.2$** | **$85.2 \pm 0.4$** | **$50.2 \pm 1.0$** | **$88.1 \pm 0.4$** | **$56.3 \pm 1.0$** |
| | Gain | +1.0 | +3.8 | +0.9 | +3.6 | +0.1 | +1.9 | +0.3 | +1.8 |

the robustness of DBGL, we adopt the same setting on all datasets, with a learning rate of 0.005 and a batch size of 256. The hidden state is set as 16, and the codebook size is set as 4096. The layers of EdgeSAGE are set to 2. All experiments are conducted on a NVIDIA GeForce RTX 4090 GPU. We conduct five repeated experiments with different random seeds on each task for evaluation. Since all tasks are binary classification, we adopt AUC-PRC and AUC-ROC as the evaluated metrics.

## 4.2 MAIN RESULTS

**Classic time series classification.** We first evaluate DBGL on IMTS classification tasks across the four IMTS datasets. This setup allows us to compare DBGL against both non-graphical and graph-based baselines under conventional evaluation metrics, highlighting its effectiveness in modeling irregularly sampled clinical data. The results are shown in the Table 2.

From the results, several key observations emerge. First, DBGL consistently outperforms all baselines across datasets, achieving the highest AUROC and AUPRC in every case. Compared to the strongest graph-based competitor (KEDGN variants), DBGL yields improvements of up to +3.8% in AUPRC and +1.0% in AUROC, demonstrating its superior ability to capture variable dependencies and temporal irregularities. Second, while non-graphical methods such as GRU-D, ODE-RNN, and IP-Net benefit from sequential modeling, they generally lag behind graph-based approaches, highlighting the importance of explicitly modeling inter-variable relations in clinical time series. Third, existing graph-based methods, including MTGNN, Raindrop, and KEDGN variants, achieve strong performance by capturing spatial-temporal dependencies, yet they still fall short of DBGL, likely due to their limited handling of irregular sampling and variable-specific decay patterns.

Overall, these results validate that the bipartite graph construction in DBGL, together with its node-specific decay mechanism, effectively models the inherent irregularities in medical time series, leading to more accurate and robust representations for downstream prediction tasks.

**Leave-variables-out.** To better verify the robustness of DBGL, we also conduct the Leave-variables-out experiments as in previous work (Luo et al. (2024)), which test the model when a subset of variables is completely missing. The discarding rate of each variable is set from 10% to 50% with a step of 10%, and all their observations in both validation and test sets will also be hidden. The results, compared with baselines on the P12 dataset, are shown in Table 3.

DBGL consistently outperforms all competitors across all discard rates. Even when up to 50% of variables are missing, DBGL maintains strong predictive performance (AUROC 81.3%, AUPRC 42.7%), substantially higher than the best baseline (KEDGN) by +4.4% AUROC and +3.5% AUPRC at the highest discard level. The performance gap increases as more variables are discarded, indicating that DBGL's bipartite graph structure and node-specific decay mechanism effectively propagate information from available variables and mitigate information loss. In contrast, both sequen-

Table 3: Performance comparison (AUROC & AUPRC, %) under different variable discard ratios on P12 dataset. The best results are highlighted in **bold** and the second-best results are in underlined.

| Method | 10% | | 20% | | 30% | | 40% | | 50% | |
|---|---|---|---|---|---|---|---|---|---|---|
| | AUROC | AUPRC | AUROC | AUPRC | AUROC | AUPRC | AUROC | AUPRC | AUROC | AUPRC |
| GRU-D | $68.6 \pm 2.3$ | $35.8 \pm 2.2$ | $68.2 \pm 2.1$ | $34.5 \pm 2.9$ | $66.8 \pm 3.3$ | $32.7 \pm 4.6$ | $65.8 \pm 4.0$ | $31.3 \pm 5.2$ | $65.1 \pm 4.1$ | $30.4 \pm 5.5$ |
| mTAND | $74.9 \pm 0.6$ | $37.7 \pm 0.6$ | $74.0 \pm 1.3$ | $36.5 \pm 1.5$ | $71.4 \pm 3.8$ | $34.1 \pm 3.7$ | $70.6 \pm 3.6$ | $33.2 \pm 3.7$ | $70.1 \pm 3.5$ | $32.5 \pm 3.6$ |
| DGM$^2$-O | $76.3 \pm 1.1$ | $39.3 \pm 1.5$ | $76.1 \pm 1.1$ | $38.2 \pm 1.7$ | $74.8 \pm 2.2$ | $36.8 \pm 2.6$ | $72.0 \pm 5.3$ | $34.3 \pm 5.0$ | $70.4 \pm 5.9$ | $32.7 \pm 5.7$ |
| MTGNN | $71.2 \pm 2.1$ | $30.5 \pm 1.5$ | $70.3 \pm 3.3$ | $29.7 \pm 2.8$ | $68.9 \pm 4.2$ | $28.5 \pm 3.3$ | $68.1 \pm 4.7$ | $27.7 \pm 3.6$ | $67.6 \pm 5.2$ | $27.2 \pm 3.8$ |
| Raindrop | $73.2 \pm 1.6$ | $32.4 \pm 0.9$ | $73.0 \pm 1.6$ | $31.7 \pm 1.1$ | $72.2 \pm 2.6$ | $31.1 \pm 2.7$ | $71.5 \pm 3.5$ | $30.6 \pm 3.5$ | $70.8 \pm 4.2$ | $29.7 \pm 4.3$ |
| DuETT | $73.9 \pm 1.7$ | $35.8 \pm 2.3$ | $74.7 \pm 1.8$ | $35.3 \pm 2.0$ | $73.6 \pm 2.2$ | $34.1 \pm 2.4$ | $72.8 \pm 2.6$ | $33.3 \pm 2.7$ | $72.3 \pm 2.7$ | $32.6 \pm 2.8$ |
| Warpformer | $75.9 \pm 0.7$ | $37.3 \pm 2.2$ | $75.6 \pm 0.8$ | $36.7 \pm 2.3$ | $73.8 \pm 2.9$ | $34.3 \pm 4.1$ | $72.8 \pm 3.4$ | $33.0 \pm 4.6$ | $72.1 \pm 3.7$ | $32.2 \pm 4.7$ |
| KEDGN | $79.7 \pm 0.4$ | $43.6 \pm 1.2$ | $79.2 \pm 0.8$ | $42.5 \pm 1.6$ | $77.7 \pm 2.2$ | $40.0 \pm 4.0$ | $77.2 \pm 2.2$ | $39.6 \pm 3.7$ | $76.9 \pm 2.2$ | $39.2 \pm 3.5$ |
| DBGL | $\mathbf{86.3 \pm 0.8}$ | $\mathbf{52.1 \pm 1.9}$ | $\mathbf{86.0 \pm 0.7}$ | $\mathbf{51.6 \pm 2.2}$ | $\mathbf{83.9 \pm 1.2}$ | $\mathbf{48.4 \pm 2.1}$ | $\mathbf{83.5 \pm 1.6}$ | $\mathbf{47.3 \pm 3.7}$ | $\mathbf{81.3 \pm 0.8}$ | $\mathbf{42.7 \pm 1.4}$ |
| Gain | +6.6 | +8.5 | +6.8 | +9.1 | +6.2 | +8.4 | +6.3 | +7.7 | +4.4 | +3.5 |

tial (GRU-D, mTAND) and existing graph-based methods degrade significantly under high missing rates, highlighting the limitations of their imputation or message-passing strategies in handling variable sparsity. More results on other datasets can be found in the Appendix G.2

Overall, these results demonstrate that DBGL is highly robust to missing variables and can reliably model irregular clinical time series even under substantial data scarcity.

## 4.3 ANALYSIS

We also conduct an analysis of confidence in detecting positive cases from our DBGL and the best baseline KEDGN by examining the average predicted probability on positive-class samples. This metric provides a direct measure of how strongly a model supports its positive predictions, with higher values indicating stronger discriminative capability.

As shown in Figure 2, DBGL consistently yields higher positive-class probabilities than KEDGN across all four datasets. The most notable gain appears on Physionet (0.7563 vs. 0.6543, +0.1020), while consistent improvements are also observed on P19 (+0.0411), MIMIC-III (+0.0356), and P12 (+0.0454). These results suggest that DBGL learns more robust and informative representations, enabling reliable identification of positive samples even under heterogeneous data distributions, which is crucial for the clinical field.

From a clinical perspective, stronger confidence in positive cases reflects more reliable disease detection, which is critical for reducing missed diagnoses and supporting decision-making in time-critical, real-world settings such as the ICU. For example, in P19, accurate identification of patients at risk of sepsis within 6 hours

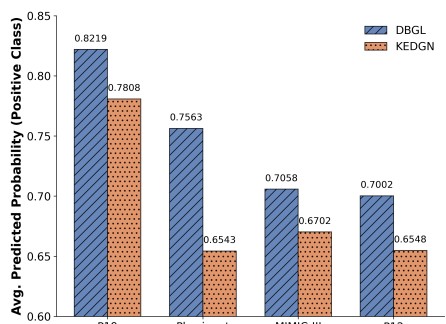

Figure 2: Comparison of Positive-class Predicted Probabilities.

enables timely interventions, while in P12, detecting patients likely to experience prolonged ICU stays informs resource allocation during the first 48 hours. Similarly, in MIMIC-III and Physionet, high-confidence predictions of in-hospital mortality allow clinicians to prioritize monitoring and treatment for high-risk patients. Across these diverse datasets, models that exhibit strong positive-case confidence provide more trustworthy and actionable clinical insights.

## 4.4 ABLATION STUDY

**Ablation Study.** We conduct ablation studies to evaluate the contribution of each key component of DBGL across all four datasets. Specifically, we design the following variants: (1) **w/o TDE**: removing the temporal decay encoding mechanism; (2) **w/o SNA**: removing the state-aware node-specific attention for patient node updates; (3) **w/o HVS**: discarding hidden variable states from the final representation; (4) **w/o CB**: removing the learnable codebook; (5) **w/o MCV**: discarding the matched code vector retrieved from the codebook for classification. (6) **w/o TE**: removing the Time

Table 4: Ablation study results (AUROC & AUPRC, %) on four clinical datasets. The best results are highlighted in **bold**.

| Method | P19 | | Physionet | | MIMIC-III | | P12 | |
|---|---|---|---|---|---|---|---|---|
| | AUROC | AUPRC | AUROC | AUPRC | AUROC | AUPRC | AUROC | AUPRC |
| w/o TDE | $92.9 \pm 0.7$ | $64.3 \pm 2.0$ | $88.7 \pm 0.8$ | $59.1 \pm 2.2$ | $84.6 \pm 1.1$ | $48.2 \pm 2.4$ | $87.2 \pm 0.5$ | $53.3 \pm 1.9$ |
| w/o SNA | $92.4 \pm 0.6$ | $65.0 \pm 1.0$ | $88.2 \pm 1.1$ | $56.7 \pm 5.1$ | $84.9 \pm 0.3$ | $49.1 \pm 1.0$ | $87.3 \pm 0.6$ | $53.3 \pm 2.3$ |
| w/o HVS | $92.3 \pm 0.5$ | $64.9 \pm 1.2$ | $87.1 \pm 2.3$ | $55.6 \pm 3.8$ | $84.5 \pm 1.8$ | $46.9 \pm 4.1$ | $86.9 \pm 0.5$ | $53.1 \pm 0.8$ |
| w/o CB | $92.8 \pm 0.5$ | $65.3 \pm 1.4$ | $88.5 \pm 1.4$ | $58.0 \pm 4.8$ | $84.9 \pm 0.3$ | $49.5 \pm 0.9$ | $87.7 \pm 0.5$ | $54.2 \pm 1.7$ |
| w/o MCV | $93.1 \pm 0.3$ | $66.1 \pm 1.4$ | $88.9 \pm 0.5$ | $59.5 \pm 1.1$ | $85.1 \pm 0.2$ | $50.0 \pm 0.2$ | $87.9 \pm 0.2$ | $54.6 \pm 0.5$ |
| w/o TE | $86.8 \pm 2.0$ | $63.7 \pm 1.7$ | $88.2 \pm 0.8$ | $58.8 \pm 2.4$ | $85.0 \pm 0.3$ | $50.1 \pm 0.3$ | $87.1 \pm 0.5$ | $53.8 \pm 1.9$ |
| Full | $\mathbf{93.3 \pm 0.5}$ | $\mathbf{66.3 \pm 1.4}$ | $\mathbf{89.1 \pm 0.3}$ | $\mathbf{60.8 \pm 2.2}$ | $\mathbf{85.2 \pm 0.4}$ | $\mathbf{50.2 \pm 1.0}$ | $\mathbf{88.1 \pm 0.4}$ | $\mathbf{56.3 \pm 1.0}$ |

Table 5: Performance comparison of different kernel types across four clinical datasets (AUROC & AUPRC, %). The best results are highlighted in **bold**.

| Kernel | P19 | | Physionet | | MIMIC-III | | P12 | |
|---|---|---|---|---|---|---|---|---|
| | AUROC | AUPRC | AUROC | AUPRC | AUROC | AUPRC | AUROC | AUPRC |
| MLP + Linear | $93.2 \pm 0.7$ | $65.6 \pm 0.9$ | $89.3 \pm 0.4$ | $60.8 \pm 2.2$ | $84.8 \pm 0.2$ | $49.5 \pm 1.0$ | $87.6 \pm 0.3$ | $54.2 \pm 0.5$ |
| MLP + Gaussian | $93.3 \pm 0.4$ | $66.1 \pm 1.6$ | $89.3 \pm 0.4$ | $60.7 \pm 2.4$ | $84.6 \pm 0.9$ | $48.3 \pm 2.5$ | $87.7 \pm 0.5$ | $54.5 \pm 1.3$ |
| MLP + Exp | $93.3 \pm 0.5$ | $66.3 \pm 1.4$ | $89.1 \pm 0.3$ | $60.8 \pm 2.2$ | $85.2 \pm 0.4$ | $50.2 \pm 1.0$ | $88.1 \pm 0.4$ | $56.3 \pm 1.0$ |
| Exp | $92.9 \pm 0.3$ | $65.2 \pm 0.6$ | $88.8 \pm 0.7$ | $58.2 \pm 1.9$ | $84.9 \pm 0.2$ | $49.9 \pm 0.5$ | $87.0 \pm 0.3$ | $53.0 \pm 1.0$ |

Embedding. These variants enable us to disentangle the effects of temporal modeling, node-specific attention, hidden state preservation, and codebook constraints on the overall performance.

Table 4 reports the ablation results on all four datasets. Several observations can be made. First, removing the temporal decay encoding mechanism (**w/o TDM**) consistently leads to a noticeable drop in both AUROC and AUPRC across datasets, highlighting the importance of explicitly modeling temporal decay in irregular sequences. Second, discarding the state-aware node-specific attention (**w/o SNA**) results in degraded performance, especially on Physionet, where the AUPRC drops sharply, confirming that patient-specific node updates are crucial for handling heterogeneous and irregular dynamics. Third, excluding hidden variable states (**w/o HVS**) yields the most pronounced performance decline, particularly in AUPRC, indicating that preserving latent states is essential for capturing long-term dependencies under irregular sampling. Fourth, removing the codebook (**w/o CB**) also impairs performance, though to a lesser extent, suggesting that the learned representation space benefits from codebook constraints. Finally, ignoring the matched code vector (**w/o MCV**) reduces both AUROC and AUPRC compared to the full model, showing the utility of codebook-guided classification. Overall, the full DBGL model achieves the best performance on all datasets, demonstrating the complementary contributions of temporal decay modeling, node-specific attention, hidden state preservation, and codebook design.

**Kernel function ablation in TDE.** To investigate the effect of alternative continuous-time kernels, we conducted additional experiments by replacing the original exponential decay: **Gaussian Kernel**: $\gamma = exp(-(\lambda \cdot \Delta t)^2)$, and **Linear Kernel**: $\gamma = max(1 - \lambda \cdot \Delta t, 0)$., where $\lambda$ is the decay rate. Both of these keep the MLP-learned decay rate. Experimental results are shown in Table 5.

The MLP + exponential kernel used in DBGL achieves the best overall performance, confirming the benefit of learning variable-specific decay rates. Alternative kernels perform slightly worse, demonstrating DBGL's robustness to kernel choice. Softplus function ensures $\lambda \geq 0$, so $exp(-\lambda * \Delta t) \in (0, 1]$, preventing overflow/underflow even for large $\Delta t$. $\gamma$ decreases monotonically with $\Delta t$ and varies with variable-specific edge attributes via the MLP, mapping variables with different temporal dynamics to distinct decay behaviors. This ensures both smooth gradients and identifiability of variable dynamics. In summary, the combination of Softplus and the exponential function guarantees a smooth, differentiable, and numerically stable decay, while effectively capturing heterogeneous temporal dynamics across variables.

### 4.5 COMPARISON TO PATIENT-TIMESTEP GRAPH.

To better verify the effect of the proposed patient-variable bipartite graph, we also conduct a variant of DBGL that constructs each variable as a patient-timestep bipartite graph. The motivation behind this comparison is that the patient-timestep bipartite graph, where the emphasis is placed on temporal alignment across different measurements. The experimental results on four datasets are shown in Figure 3. We regard our DBGL as PVG (patient-variable graph) and the patient-timestep graph as PTG. The experimental results show that PTG has experienced an extremely significant performance degradation on each dataset.

The advantage of the patient-timestep bipartite graph lies in its intuitive depiction of patients' changes over time, emphasizing temporal alignment and dynamic trends, with a simple structure that is well-suited for modeling global temporal dependencies. However, this design may overlook the heterogeneous nature of variables and their irregular sampling characteristics, since it treats each variable instance at a given time as independent nodes without explicitly encoding variable-level semantics. By contrasting the two designs, we aim to highlight the advantage of modeling variable-level relationships under irregular sampling, which allows DBGL to better capture variable-specific dynamics

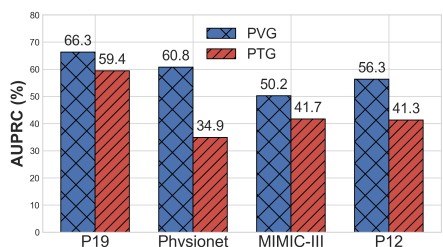

Figure 3: Comparison of patient-variable graph and patient-timestep graph.

and exploit richer cross-variable dependencies. This comparison demonstrates the necessity of our proposed DBGL and clarifies its contribution beyond existing timestep-centered formulations.

## 5 CONCLUSION

In this paper, we propose DBGL, a novel framework for learning IMTS representation and classification. DBGL directly encodes the irregular sampling pattern of each variable in each time step into a patient-variable bipartite graph, effectively capturing temporal sampling irregularities. Using this graph structure, we introduce a temporal decay encoding mechanism, which allows each variable to decay at a rate proportional to its sampling interval and observation. This design enables DBGL to utilize the unique historical state, sampling interval, and decay characteristics of each variable to perform precise state updates at every time step, resulting in more informative and discriminative representations. Extensive experiments on four publicly available clinical datasets demonstrate that DBGL consistently outperforms state-of-the-art baselines. Overall, DBGL provides a general and flexible framework for modeling irregularly sampled clinical time series, offering both improved predictive performance. In future work, we plan to extend DBGL to handle multimodal patient data, incorporate uncertainty estimation for clinical decision support, and explore its application to other time-critical healthcare tasks.

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

Table 6: Variable-specific decay rates ($\lambda$) on the P12 dataset. Higher $\lambda$ indicates faster-changing.

| Variable | $\lambda$ | Variable | $\lambda$ | Variable | $\lambda$ | Variable | $\lambda$ |
|---|---|---|---|---|---|---|---|
| ALP | 14.7113 | FiO2 | 0.2458 | NISysABP | 6.0698 | ALT | 14.6235 |
| GCS | 1.0948 | Na | 0.0420 | AST | 13.8853 | Glucose | 0.0493 |
| PaCO2 | 0.0225 | Albumin | 0.0853 | HCO3 | 0.1094 | PaO2 | 0.0280 |
| BUN | 0.0854 | HCT | 0.0379 | Platelets | 0.0505 | Bilirubin | 0.8760 |
| HR | 1.7301 | RespRate | 4.1157 | Cholesterol | 7.0000 | K | 0.0368 |
| SaO2 | 0.0408 | Creatinine | 0.1399 | Lactate | 13.0000 | SysABP | 0.1403 |
| DiasABP | 0.0232 | MAP | 0.0913 | Temp | 0.0265 | MechVent | 9.4304 |
| Mg | 0.1023 | TropI | 0.0684 | NIDiasABP | 0.0272 | NIMAP | 0.9093 |
| TropT | 14.0000 | Urine | 0.0986 | WBC | 0.0545 | pH | 10.8730 |

Table 7: Variable-specific decay rates ($\lambda$) on the P19 dataset. Higher $\lambda$ indicates faster-changing.

| Variable | $\lambda$ | Variable | $\lambda$ | Variable | $\lambda$ | Variable | $\lambda$ |
|---|---|---|---|---|---|---|---|
| Heart rate | 0.0759 | Temp | 0.1131 | BaseExcess | 14.0 | SpO2 | 0.0822 |
| SBP | 0.0813 | HCO3 | 0.0900 | MAP | 0.0820 | DBP | 0.0798 |
| FiO2 | 0.4244 | Resp | 0.0818 | EtCO2 | 0.1071 | pH | 0.0953 |
| PaCO2 | 0.7904 | SaO2 | 16.0 | AST | 16.235 | BUN | 16.0 |
| Alkalinephos | 15.0 | Calcium | 0.1144 | Chloride | 0.0782 | Creatinine | 0.1134 |
| Bilirubin_direct | 15.0 | Glucose | 0.1376 | Lactate | 3.7805 | Magnesium | 0.0850 |
| Phosphate | 0.1111 | Potassium | 0.1036 | Bilirubin_total | 16.0 | TroponinI | 0.1478 |
| Hct | 0.1004 | Hgb | 0.0914 | PTT | 0.1305 | WBC | 0.1015 |
| Fibrinogen | 0.0890 | Platelets | 0.0896 | — | — | — | — |

# APPENDIX

## A  VARIABLE DECAY ANALYSIS

We conducted a systematic analysis of the decay rates of different variables across all datasets. First, we clarify the meaning of the variable decay rate: one of the core components of our method is the "variable decay irregularity" model, which characterizes the temporal dynamics of each clinical variable—specifically, the rate at which its autocorrelation decays over time. A higher decay rate indicates stronger short-term fluctuations (e.g., heart rate and blood pressure, which typically vary on a minute-level scale), whereas a lower decay rate reflects greater temporal stability (e.g., creatinine and hemoglobin, which often change over hours or days). This metric therefore provides a unified quantitative way to capture clinically meaningful differences in temporal behavior across variables.

To further deal with the concern regarding whether different clinical variables exhibit heterogeneous temporal decay, we conducted an autocorrelation-based analysis on all training samples. For each variable, we computed the empirical autocorrelation over time lags ($\Delta t$) and fitted an exponential model $\text{Corr}(\Delta t) = e^{-\lambda \Delta t}$ to estimate its decay rate.

Similar patterns are observed in P19, Physionet, and MIMIC-III. We also provided Kruskal–Wallis test results as Table 10. Kruskal–Wallis tests across all datasets indicate that variable-specific decay rates ($\lambda$) differ significantly (all $p \ll 0.05$). This confirms the presence of *variable decay irregularity*, i.e., different clinical variables exhibit distinct temporal dynamics. For example, high-$\lambda$ variables such as Lactate and ALP change rapidly over time, whereas low-$\lambda$ variables like Creatinine and Albumin are relatively stable. These differences suggest that predictive models must account for variable-specific temporal behaviors to achieve robust clinical predictions. Notably, the MIMIC-III dataset shows the largest test statistic, indicating the strongest heterogeneity in decay rates among its variables.

This observed heterogeneity underscores the necessity of variable-specific decay modeling. Using a uniform or hand-crafted decay assumption would fail to capture these differences, effectively treating both stable indicators and highly reactive biomarkers as if they shared the same temporal

Table 8: Variable-specific decay rates ($\lambda$) on the Physionet dataset. Higher $\lambda$ indicates faster-changing.

| Variable | $\lambda$ | Variable | $\lambda$ | Variable | $\lambda$ | Variable | $\lambda$ |
|---|---|---|---|---|---|---|---|
| ALP | 16.5173 | ALT | 14.4846 | AST | 16.0 | Albumin | 0.1164 |
| BUN | 0.0400 | Bilirubin | 14.0 | Cholesterol | 2.0 | Creatinine | 0.0502 |
| DiasABP | 0.0212 | FiO2 | 0.3273 | GCS | 13.5627 | Glucose | 0.0586 |
| HCO3 | 0.0791 | HCT | 0.0392 | HR | 12.0 | K | 0.0429 |
| Lactate | 12.8876 | MAP | 0.0371 | MechVent | 9.4304 | Mg | 14.0 |
| NIDiasABP | 0.0272 | NIMAP | 1.0679 | NISysABP | 11.0 | Na | 0.0441 |
| PaCO2 | 0.0289 | PaO2 | 0.0326 | Platelets | 0.0721 | RespRate | 3.3877 |
| SaO2 | 0.2059 | SysABP | 0.1289 | Temp | 0.0435 | TropI | 0.1035 |
| TropT | 14.0 | Urine | 0.0279 | WBC | 0.1009 | pH | 11.9039 |

Table 9: Variable-specific decay rates ($\lambda$) on the MIMIC-III dataset. Higher $\lambda$ indicates faster-changing.

| Variable | $\lambda$ | Variable | $\lambda$ | Variable | $\lambda$ | Variable | $\lambda$ |
|---|---|---|---|---|---|---|---|
| Weight | 6098.3438 | HR | 11.8839 | MAP | 138.8822 | DBP | 4277.9697 |
| SBP | 581.1183 | SpO2 | 2053.2156 | RR | 11829.0942 | CRR | 517.8212 |
| Glucose | 17.7346 | pH | 182.4216 | Temperature | 163.4556 | FiO2 | 18.1914 |
| GCS-EO | 14.107 | GCS-MR | 26.0087 | GCS-T | 10.5168 | GCS-VR | 15.660 |

patterns. By employing our designed variable decay encoding module, the proposed method can adaptively capture these irregular dynamics, enabling a more accurate representation of clinical time series and improving performance on downstream tasks.

## B    ALGORITHM OF DBGL

We provide the pseudo-code of DBGL here as a better illustration.

---
**Algorithm 1** The pseudo-code of DBGL.

---
**Require:** Input IMTS $X$, sampling timestamps $\mathbf{T}$, binary mask matrix $M$, codebook size $K$
**Ensure:** Predicted labels $\hat{\mathbf{y}}$
 1: Extract latent embedding $X'$ of input IMTS $X$
 2: Initialize patient-variant bipartite graphs $\mathcal{G}$ from $X'$ and $M$
 3: **for** each time step $t$ in $X$ **do**
 4:     **if** $t > 1$ **then**
 5:         Conduct codebook soft-fusion as Eq. 7 and Eq. 8
 6:         Aggregate hidden states to patient nodes for time-step representation $h^t$
 7:     **end if**
 8:     **for** each variable $v$ in $G$ **do**
 9:         Obtain the edge embedding $E$ from $X'$
10:         Pass message over the graph and aggregation as Eq. 1 and Eq. 2.
11:         Compute node-specific decay: $h_v^t \leftarrow h_v^{t-1} \cdot \exp(-\gamma_v \Delta t)$
12:         Update hidden state with current observation (edge feature) $e_{p,n}^t$ as Eq. 4
13:     **end for**
14: **end for**
15: Obtain matched codebook representation as Eq. 10
16: Concatenate learned features for $\mathbf{z}_p$ and predict labels $\hat{y}_p = MLP(\mathbf{z}_p)$ as Eq. 11.
17: **return** $\hat{y}_p$

---

Table 10: Kruskal–Wallis test results for variable-specific decay rates ($\lambda$) across datasets.

| Dataset | Statistic | p-value |
|---------|-----------|---------|
| P12 | 668.4338 | 3.87e-118 |
| P19 | 446.4491 | 5.97e-74 |
| Physionet | 267.0967 | 5.56e-38 |
| MIMIC-III | 697.995 | 4.97e-139 |

## C  DATASET DETAILS

Four public irregular medical time series datasets are used for evaluation. The statistics of these datasets are summarized in Table 11.

Table 11: Dataset statistics

| Datasets | # Samples | # Variables | # Max Length | Missing Ratio | Task |
|----------|-----------|-------------|--------------|---------------|------|
| P19 | 38803 | 34 | 60 | 94.9% | Sepsis Prediction |
| P12 | 11988 | 36 | 215 | 88.4% | Stay Predcition |
| MIMIC-III | 21107 | 16 | 292 | 65.5% | Mortality Prediction |
| Physionet | 3997 | 36 | 215 | 84.9% | Mortality Prediction |

**P19 Dataset**. Originating from the PhysioNet 2019 Sepsis Early Prediction Challenge (Reyna et al. (2020)), this dataset comprises medical records of 38,803 patients. All samples are associated with a binary classification label indicating whether sepsis occurred within the subsequent 6 hours. Each record includes 34 time-series variables (with a maximum time span of 60 hours) and a static feature vector containing attributes such as age, gender, time from hospital admission to ICU admission, ICU type, and length of stay in days. Samples with abnormal time-series lengths were excluded following the criteria outlined in Zhang et al. (2022). The dataset is available at: https://physionet.org/content/challenge-2019/1.0.0/. This dataset performs patient-level data splitting, wtih 8:1:1 for training, validation, and test set.

**P12.** After removing 12 invalid samples identified in Horn et al. (2020), the P12 dataset [46] contains 11,988 valid patient records. The binary labels are determined based on the length of ICU stay: a negative label indicates a short stay ($\leq 3$ days), and a positive label indicates a long stay ($> 3$ days). Each patient's record includes multivariate time series collected from 36 types of sensors (excluding weight measurements) during the first 48 hours of ICU admission, along with a 9-dimensional static feature vector containing demographic and other relevant attributes. The data can be accessed at: https://physionet.org/content/challenge-2012/1.0.0/. This dataset performs patient-level data splitting, wtih 8:1:1 for training, validation, and test set.

**MIMIC-III.** MIMIC-III (Johnson et al. (2016)) is a widely used public medical database containing de-identified electronic health records of ICU patients at the Beth Israel Deaconess Medical Center between 2001 and 2012. This study employs its binary in-hospital mortality prediction task for evaluation. The original data includes approximately 57,000 ICU records, covering multidimensional variables such as medication records, in-hospital mortality, and vital signs. After preprocessing, 21,107 samples were obtained, each containing 16 clinical features over 48 hours. The dataset is available at: https://physionet.org/content/mimiciii/1.4/. This dataset performs record-level data splitting, wtih 70%/15%/15% for training, validation, and test set.

**Physionet.** Physionet Silva et al. (2012) contains monitoring data from the first 48 hours of ICU patient admissions. This study primarily uses the in-hospital mortality prediction task. Following the same preprocessing pipeline as applied to the P12 dataset, 3,997 annotated samples were obtained. The data can be found at: https://physionet.org/content/challenge-2012/. This dataset performs patient-level data splitting, wtih 8:1:1 for training, validation, and test set.

## D    EVALUATED METRICS

**AUROC.** In irregular clinical time-series classification, AUROC measures the model's overall ability to distinguish positive from negative samples across all thresholds, reflecting its ranking or discrimination power. Even under sparse or unevenly sampled observations, AUROC provides a stable assessment of how well the model separates positive and negative cases.

**AUPRC.** In contrast, AUPRC focuses on the precision-recall trade-off, which is especially critical in clinical datasets where positive events are rare. High AUPRC indicates that the model can capture most true positive events while keeping false positives low. In irregular and imbalanced medical time-series, accurate identification of these rare but clinically important events is often more consequential than overall ranking, making AUPRC a more relevant metric than AUROC for evaluating predictive performance in such settings.

## E    BASELINES

Following Luo et al. (2024), all baseline models are implemented according to the descriptions in their original papers or the default configurations of their official code repositories. The details are as follows:

GRU-D (Che et al. (2018)): Based on the advanced Gated Recurrent Unit (GRU), this method incorporates two representations of missing patterns—masking and time intervals—and seamlessly integrates them into the model architecture. Code: https://github.com/Han-JD/GRU-D.

ODE-RNN (Rubanova et al. (2019)): This model uses neural ordinary differential equations (ODEs) to model hidden state dynamics and employs an RNN to update the hidden state when new observations are encountered. Code: https://github.com/YuliaRubanova/latent_ode.

IP-Net (Shukla & Marlin (2019)): A model based on a multi-layer semi-parametric interpolation structure, which processes irregular sequences using an interpolation network followed by a predictive GRU network. Code: https://github.com/mlds-lab/interp-net.

SeFT (Horn et al. (2020)): This approach uses a set function method where each observation is modeled independently before being aggregated via an attention mechanism. Code: https://github.com/BorgwardtLab/SeFT.

mTAND (Shukla & Marlin (2021b)): A deep learning framework designed for irregular multivariate time series, which employs continuous-time embedding learning and an attention mechanism to produce fixed-length representations. Code: https://github.com/reml-lab/mTAN.

DGM$^2$-O (Wu et al. (2021)): A generative model that captures the evolution of latent clusters rather than independent feature representations, enabling robust modeling of sparse time series. Code: https://github.com/thuwuyinjun/DGM2.

StraTS (Tipirneni & Reddy (2022)): A self-supervised Transformer model for sparse and irregular multivariate time series. Code: https://github.com/sindhura97/STraTS.

DuETT (Labach et al. (2023)): A dual-event time Transformer model designed for Electronic Health Records (EHRs). Code: https://github.com/layer6ai-labs/DuETT.

ViTST (Li et al. (2023)): This method converts irregular multivariate time series into line graph images and adapts vision Transformer models to handle time series classificatioliketo image classification. Code: https://github.com/Leezekun/ViTST.

Warpformer (Zhang et al. (2023)): A Transformer-based network that extracts multi-scale features using deformable warping modules and a dual-attention mechanism. Code: https://github.com/imJiawen/Warpformer.

MTGNN (Wu et al. (2020)): A general graph neural network framework specifically designed for multivariate time series. Code: https://github.com/nnzhan/MTGNN

Raindrop (Zhang et al. (2021)): A graph neural network-based model that learns sensor dynamics purely from observed data. Code: https://github.com/mimsharvard/Raindrop

ISIM (Chen et al. (2025b)): A novel method to model the irregular time series data as an image and incorporate both sequence and image representations for a more generalizable joint representation. Since there was no code, we directly listed the results in the paper.

TimeCHEAT (Liu et al. (2025)): A method to use the channel-dependent strategy locally and the channel-independent strategy globally for better learning of irregularly multivariate time series. Since there was no code, we directly listed the results in the paper.

KEDGN (Luo et al. (2024)): A method to use a pretrained language model for semantic representation extraction of each variable from the textual medical knowledge. Code:https://github.com/qianlima-lab/KEDGN.

# F COMPUTATIONAL COSTS AND BOTTLENECK ANALYSIS

## F.1 COMPARISON OF COMPUTATIONAL COSTS.

We analyzes of the time and space overhead on the Physionet dataset, with a batch size of 128, following Luo et al. (2024). We compare the training time per epoch (min/epoch) and the used space on GPU (MiB) as shown in Table 12.

As shown in the results, our proposed DBGL achieves the highest average AUPRC (60.8% ± 2.2%) on the Physionet dataset, significantly outperforming all baseline models and demonstrating its superior ability to distinguish between positive and negative samples. Despite its strong performance, DBGL maintains a moderate training time per epoch (0.52 min) and reasonable GPU memory usage (2674 MiB), indicating an effective trade-off between predictive accuracy and computational efficiency.

In comparison, sequential models such as ODE-RNN and GRU-D exhibit longer

Table 12: Training time and memory consumption per epoch of different models.

| Model | Time (min/epoch) | Space (MiB) | AUPRC (%) |
|---|---|---|---|
| ODE-RNN | 5.06 | 2582 | 33.7 ± 4.1 |
| GRU-D | 1.32 | 796 | 42.7 ± 7.2 |
| SeFT | 0.07 | 684 | 29.4 ± 0.9 |
| mTAND | 0.05 | 4658 | 52.5 ± 1.3 |
| DGM$^2$-O | 0.06 | 684 | 50.4 ± 3.2 |
| Raindrop | 0.17 | 4864 | 41.2 ± 3.6 |
| Warpformer | 0.33 | 11084 | 43.5 ± 2.3 |
| KEDGN | 0.44 | 1798 | 57.5 ± 2.5 |
| DBGL | 0.52 | 2674 | 60.8 ± 2.2 |

training times with limited predictive performance (AUPRC of 33.7% and 42.7%, respectively). Lightweight models like SeFT and DGM$^2$-O offer low computational overhead but achieve lower AUPRC values. Transformer-based models such as Warpformer achieve moderate performance (43.5%) but incur substantially higher memory costs (11084 MiB), limiting their scalability. Other graph-based models, including KEDGN and mTAND, perform well but still fall short of DBGL in AUPRC. In addition, KEDGN also requires the pre-trained language model to be used in advance for variable semantic extraction, which is an additional computational overhead. Overall, these results demonstrate that DBGL provides state-of-the-art predictive performance while maintaining computational efficiency, making it highly suitable for large-scale clinical datasets and practical deployment scenarios.

## F.2 BOTTLENECK FOR SCALING

The proposed patient–variable bipartite graph is constructed on a per-batch basis. Specifically, we conducted experiments on a single 24GB NVIDIA GeForce RTX 4090 GPU using the P12 dataset, exploring the impact of batch size (B), codebook size (C), sequence length (L), and number of variables (V). To vary L and V, we repeated sequences and variables accordingly to increase sequence length and variable count. We report both GPU memory consumption and training time per epoch. Our baseline configuration is B=256, V=36, L=215, C=4096, and when varying one parameter, the others are kept constant.

The results show that increasing batch size B significantly increases GPU memory usage (e.g., from 2670 MiB at B=128 to 19700 MiB at B=2048), while training time per epoch decreases due to higher GPU utilization at larger batch sizes. Increasing the number of variables V also substantially

Table 13: GPU memory usage and training time per epoch under different settings (B=batch size, V=number of variables, L=sequence length, C=codebook size).

| B | Space (MiB) | Time (s/epoch) | V | Space (MiB) | Time (s/epoch) | L | Space (MiB) | Time (s/epoch) | C | Space (MiB) | Time (s/epoch) |
|---|---|---|---|---|---|---|---|---|---|---|---|
| 128 | 2670 | 56 | 36 | 3806 | 30 | 215 | 3806 | 30 | 512 | 2950 | 28 |
| 256 | 3806 | 30 | 72 | 5258 | 35 | 430 | 6294 | 66 | 1024 | 3064 | 28 |
| 512 | 6034 | 18 | 144 | 8154 | 48 | 860 | 11254 | 192 | 2048 | 3238 | 29 |
| 1024 | 10776 | 12 | 288 | 14208 | 71 | 1290 | 16316 | 338 | 4096 | 3806 | 30 |
| 2048 | 19700 | 9 | 432 | 20306 | 89 | 1720 | 21288 | 525 | 8172 | 4968 | 31 |

increases memory usage (from 3806 MiB at V=36 to 20306 MiB at V=432) and training time (from 30 s to 89 s per epoch). The sequence length L has the most pronounced effect on both memory and training time; for example, increasing L from 215 to 1720 raises memory from 3806 MiB to 21288 MiB and epoch time from 30 s to 525 s. In contrast, codebook size C has a relatively minor impact on memory and training time (e.g., increasing C from 512 to 8172 only increases memory from 2950 MiB to 4968 MiB, with negligible change in epoch time).

Based on these results, we analyze potential scalability bottlenecks:

- GPU memory limitation is the primary constraint. Increasing batch size, number of variables, or sequence length quickly consumes GPU memory, especially in large-scale graphs or long-sequence scenarios, potentially exceeding hardware limits.

- Training time grows nonlinearly with sequence length; for very long sequences, per-epoch time may reach several hundred seconds, affecting experimental efficiency.

- In contrast, codebook size C has minimal effect on scalability, making it easier to increase without major overhead.

### F.3 COMPLEXITY ANALYSIS

**Time Complexity.** For an input of size $O(B \times T \times V \times D)$, where $B$ is the batch size, $T$ is the number of time steps, $V$ is the number of variable nodes, and $D$ is the hidden dimension, the per-step operations include:

- Codebook aggregation: $O((B + V) \cdot D \cdot N)$,
- GNN message passing (EdgeSAGEConv + edge update): $O(L \cdot E \cdot (D + E))$,
- Decay computation and gated update: $O(B \cdot V \cdot D)$,

where $L$ is the number of GNN layers, $E$ is the number of edges (smaller than $B \cdot V$ for irregular time series), and $N$ is the codebook size. Iterating over $T$ time steps gives the total time complexity:

$$O\Big(T \cdot \big(L \cdot E \cdot (D + E) + B \cdot V \cdot D + (B + V) \cdot D \cdot N\big)\Big).$$

While our method introduces additional overhead via codebook aggregation and EdgeSAGEConv, both GNN message passing and node decay updates are linear in $BVD$ and can be efficiently parallelized on GPUs.

**Space Complexity.** The memory cost for storing inputs, hidden states, codebook, and edge attributes is

$$O(B \cdot T \cdot V \cdot D + E \cdot D + N \cdot D).$$

### F.4 RUNTIME COMPARISON

To quantify the computational overhead, we compare our model (DBGL) with the sequence-based baseline KEDGN using the same GPU (a NVIDIA A800 80GB PCIe). The measured testing time (in seconds) is summarized in Table 14 and Table 15.

Several observations can be drawn from the results:

Table 14: Inference time (seconds) of DBGL and KEDGN on four datasets.

| Model | P19 | Physionet | MIMIC-III | P12 |
|-------|-----|-----------|-----------|-----|
| KEDGN | 0.11 | 0.14 | 0.61 | 0.29 |
| DBGL | 0.18 | 0.16 | 0.84 | 0.50 |

Table 15: Training time per batch (seconds) of DBGL and KEDGN on four datasets.

| Model | P19 | Physionet | MIMIC-III | P12 |
|-------|-----|-----------|-----------|-----|
| KEDGN | 1.76 | 0.93 | 5.92 | 1.61 |
| DBGL | 2.03 | 0.81 | 4.86 | 1.92 |

- The inference time of DBGL is highly comparable to that of the sequence-based KEDGN. On P19 and P12, DBGL is slightly slower (+0.27s and +0.31s), reflecting the lightweight online graph construction. On Physionet and MIMIC-III, DBGL is even faster ($-0.12$ s and $-1.06$ s), showing that dynamic graph generation does not introduce significant computational burden.

- The dynamic graph building process adds only minor overhead because graph nodes correspond directly to observed variables at each step, and the graph size remains small (patient node + variable nodes), resulting in linear complexity with respect to observed variables.

- Across all datasets, the runtime difference between DBGL and sequence baselines is within 0.3–1.0 seconds, demonstrating that the proposed graph mechanism is both practical and efficient for real-world healthcare scenarios.

In summary, although our patient–variable graphs are dynamically generated at inference time, this process introduces minimal overhead, and the overall inference speed remains competitive and often even faster than sequence-based baselines.

## G  MORE EXPERIMENTS

### G.1  DIFFERENT GROUP PATIENT ANALYSIS

We conducted extensive experiments across four datasets: P12, P19, Physionet, and MIMIC-III, each with distinct patient distributions, observation lengths, variable sets, and clinical characteristics. To examine the effect of patient cohort composition, we randomly split the training set of each dataset into two non-overlapping subsets (50% of patients each), trained separate models on each subset, and evaluated their performance. Results are shown in Table 16.

We found that performance gaps between models trained on different subsets are minimal, demonstrating strong generalization across patient distributions. Even with only 50% of the training data, our model still outperforms nearly all baselines and is only slightly below KEDGN, supporting the robustness and effectiveness of our design.

Since P12, P19, and Physionet include ICU type labels, we further evaluated model performance across different patient groups.

Across different ICU types and datasets, the model maintains strong predictive performance, demonstrating good generalization across diverse patient populations. Extremely high AUROC values often occur in ICU types with small sample sizes or highly homogeneous patient characteristics (e.g., cardiac surgery recovery patients), where clinical patterns are more consistent. While performance is stable, small-sample groups may introduce higher variance.

Overall, these results show that our model is robust, generalizes well across heterogeneous patient cohorts, and performs particularly strongly in clinically consistent populations.

Table 16: Performance on different training subsets across four clinical datasets (AUROC & AUPRC, %).

| Subset | P19 | | Physionet | | MIMIC-III | | P12 | |
|---|---|---|---|---|---|---|---|---|
| | AUROC | AUPRC | AUROC | AUPRC | AUROC | AUPRC | AUROC | AUPRC |
| Subset 1 | $90.8 \pm 0.3$ | $61.0 \pm 1.6$ | $87.0 \pm 1.5$ | $55.1 \pm 4.5$ | $83.7 \pm 0.5$ | $46.1 \pm 1.2$ | $85.9 \pm 0.6$ | $50.8 \pm 1.0$ |
| Subset 2 | $91.0 \pm 0.2$ | $61.6 \pm 1.2$ | $86.5 \pm 2.6$ | $54.6 \pm 3.3$ | $84.0 \pm 0.3$ | $45.8 \pm 1.3$ | $85.7 \pm 0.7$ | $50.8 \pm 1.9$ |
| All | $93.3 \pm 0.5$ | $66.3 \pm 1.4$ | $89.1 \pm 0.3$ | $60.8 \pm 2.2$ | $85.2 \pm 0.4$ | $50.2 \pm 1.0$ | $88.1 \pm 0.4$ | $56.3 \pm 1.0$ |

Table 17: P19 dataset: performance across ICU types.

| ICU Type | Samples | AUROC |
|---|---|---|
| Medical ICU | 3872 | $93.2 \pm 0.5$ |
| SIC Surgical ICU | 9 | $98.6 \pm 2.9$ |

### G.2 EXPERIMENTS ABOUT LEAVE-VARIABLES-OUT.

We provide more results about Leave-Variables-Out experiments on MIMIC-III, P19, and Physionet datasets here. The experimental results are shown in Table 20.

As shown in Table 20, DBGL consistently outperforms all baselines across different discarding ratios and datasets. On MIMIC-III, even when 50% of the variables are removed, DBGL maintains an AUROC of 81.1% and an AUPRC of 42.9%, significantly higher than the second-best method KEDGN (AUROC 80.0%, AUPRC 42.4%). Similar trends are observed on P19, where DBGL achieves AUROC/AUPRC scores of 90.4%/58.3% under 50% variable removal, outperforming Warpformer and KEDGN. On Physionet, DBGL also demonstrates superior robustness, with AUROC/AUPRC remaining at 82.8%/46.7% under the most extreme 50% variable drop. These results highlight the effectiveness of DBGL in leveraging patient-variant bipartite graphs and node-specific decay mechanisms to capture temporal dynamics, even under significant feature missingness. The consistent performance gain indicates that DBGL can robustly model irregularly sampled clinical time series and is less sensitive to missing or discarded variables compared to existing state-of-the-art methods.

### G.3 EFFECT OF THE NUMBER OF EDGESAGE LAYERS.

To investigate the impact of the number of EdgeSAGE layers in DBGL, we vary the number of layers from 1 to 5 in increments of 1. We also analyze the influence of graph convolution depth, as increasing the number of GCN layers allows the model to capture higher-order dependencies between patient and variable nodes, but may also introduce over-smoothing if too deep, and bring more computation cost. The experimental results are shown in Figure 4.

Figure 4 shows the AUPRC of DBGL as the number of EdgeSAGE layers varies from 1 to 5. Figure 4 shows the AUPRC of DBGL as the number of EdgeSAGE layers varies from 1 to 5. Increasing the depth from 1 to 2 consistently improves AUPRC across all datasets (e.g., P19: 65.0% → 66.3%), indicating that capturing higher-order patient-variable dependencies benefits positive-sample discrimination. Beyond 2 layers, gains plateau or fluctuate slightly, suggesting diminishing returns and potential over-smoothing. Based on these results, we select a 2-layer EdgeSAGE model as the final configuration. Beyond 2 layers, gains plateau or fluctuate slightly, suggesting diminishing returns and potential over-smoothing, and also bring a larger computational cost. Based on these results, we select a 2-layer EdgeSAGE model as the final configuration.

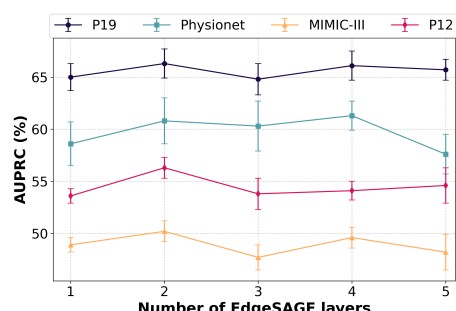

Figure 4: Effect of the number of EdgeSAGE layers.

Table 18: P12 dataset: performance across ICU types.

| ICU Type | Coronary Care Unit | Cardiac Surgery Recovery Unit | Medical ICU | SIC Surgical ICU |
|---|---|---|---|---|
| Samples | 219 | 231 | 419 | 339 |
| AUROC | 81.9±1.5 | 83.4±1.9 | 85.0±0.5 | 90.4±0.3 |

Table 19: Physionet dataset: performance across ICU types.

| ICU Type | Coronary Care Unit | Cardiac Surgery Recovery Unit | Medical ICU | SIC Surgical ICU |
|---|---|---|---|---|
| Samples | 64 | 91 | 153 | 91 |
| AUROC | 90.3±2.4 | 99.4±0.6 | 84.8±0.5 | 85.4±1.2 |

### G.4 EFFECT OF THE SIZE OF CODEBOOK.

To investigate the impact of codebook size on model performance, we vary the number of discrete codewords in the codebook and evaluate the resulting predictive accuracy and representation quality. A larger codebook allows the model to capture finer-grained variations in the input space, potentially improving expressiveness, but it may also increase computational cost and the risk of overfitting. Conversely, a smaller codebook enforces stronger quantization, which can act as a regularizer but may lead to information loss. The size of the codebook is set from 512 to 8192 and measured on all datasets.

As shown in Figure 5, increasing the codebook size generally improves performance initially, with the model achieving the highest AUROC and AUPRC on most datasets around 2048–4096 codewords. For instance, on P19, AUROC increases from 92.9% to 93.3% and AUPRC from 64.9% to 66.3% as the codebook grows from 512 to 4096. Similar trends are observed on P12 and MIMIC-III, while Physionet shows a slight drop beyond 2048, suggesting diminishing returns with overly large codebooks. These results indicate a trade-off between expressiveness and overfitting: a larger codebook enables finer-grained representation of input features, improving predictive power, but excessively large codebooks can introduce noise and marginally degrade performance.

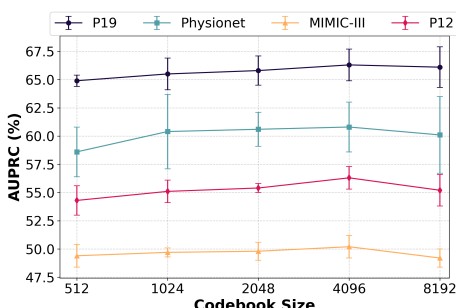

Figure 5: Comparison of Positive-class Predicted Probabilities.

To further examine codebook usage, we compute the *soft utilization rate*, defined as the fraction of code vectors receiving above-uniform activation.

Given the soft assignment weights $w \in \mathbb{R}^{N \times K}$, the utilization rate is defined as:

$$\bar{w}_k = \frac{1}{N} \sum_{i=1}^{N} w_{i,k}, \qquad \text{Utilization} = \frac{1}{K} \sum_{k=1}^{K} \mathbb{1}\!\!\mathbb{K}\left(\bar{w}_k > \frac{1}{K}\right), \qquad (12)$$

where $N$ is the number of samples and $K$ is the number of codebook entries. This metric quantifies how many codebook entries are used more frequently than what a uniform distribution would predict.

There are some key observations from the results:

- **No collapse occurs**: utilization remains $\sim 0.498$ for all datasets, far from zero.
- **Balanced engagement**: code vectors are activated evenly across the codebook, showing stable and effective use.
- **Consistent usage across datasets**: despite differing dynamics, the codebook contributes meaningfully, particularly in homogeneous datasets where similarity-guided aggregation is more beneficial.

Table 20: Performance comparison (AUROC & AUPRC, %) under different discarding ratios on three clinical datasets. The best results are highlighted in **bold** and the second-best results are in underlined.

| | Method | 10% | | 20% | | 30% | | 40% | | 50% | |
|---|---|---|---|---|---|---|---|---|---|---|---|
| | | AUROC | AUPRC | AUROC | AUPRC | AUROC | AUPRC | AUROC | AUPRC | AUROC | AUPRC |
| MIMIC-III | GRU-D | 81.0 ± 0.6 | 42.1 ± 0.8 | 80.3 ± 0.9 | 41.7 ± 1.0 | 79.2 ± 1.8 | 41.0 ± 1.4 | 78.5 ± 2.1 | 40.4 ± 1.6 | 77.9 ± 2.2 | 39.9 ± 1.8 |
| | mTAND | 81.2 ± 0.2 | 42.1 ± 0.8 | 80.4 ± 1.1 | 41.9 ± 1.2 | 79.7 ± 1.4 | 41.0 ± 1.7 | 79.3 ± 1.4 | 40.4 ± 2.0 | 78.8 ± 1.6 | 39.8 ± 2.3 |
| | DGM²-O | 78.8 ± 0.5 | 34.2 ± 0.9 | 78.3 ± 0.8 | 33.9 ± 1.1 | 77.6 ± 1.2 | 33.4 ± 1.2 | 77.3 ± 1.3 | 33.1 ± 1.2 | 76.8 ± 1.5 | 32.6 ± 1.4 |
| | MTGNN | 78.8 ± 1.1 | 34.5 ± 1.4 | 78.0 ± 1.6 | 34.0 ± 1.3 | 77.1 ± 2.2 | 33.5 ± 1.5 | 76.3 ± 2.5 | 32.8 ± 1.9 | 75.6 ± 3.2 | 32.2 ± 2.4 |
| | Raindrop | 78.2 ± 1.1 | 33.7 ± 0.9 | 77.5 ± 1.3 | 33.5 ± 0.9 | 76.4 ± 2.1 | 32.8 ± 1.4 | 76.0 ± 2.0 | 32.5 ± 1.4 | 75.7 ± 2.0 | 32.3 ± 1.4 |
| | DuETT | 78.0 ± 0.5 | 34.0 ± 0.9 | 77.2 ± 1.0 | 33.7 ± 0.8 | 76.6 ± 1.2 | 33.3 ± 1.0 | 76.4 ± 1.2 | 33.0 ± 1.0 | 76.1 ± 1.3 | 32.6 ± 1.3 |
| | Warpformer | 82.5 ± 0.5 | 43.1 ± 0.8 | 81.7 ± 0.9 | 42.5 ± 1.2 | 81.2 ± 1.1 | 42.1 ± 1.2 | 80.6 ± 1.5 | 41.8 ± 1.3 | 80.0 ± 1.9 | 41.3 ± 1.6 |
| | KEDGN | 83.0 ± 0.7 | 44.8 ± 2.0 | 82.3 ± 1.1 | 44.4 ± 1.9 | 81.3 ± 1.9 | 43.6 ± 2.2 | 80.6 ± 2.1 | 43.0 ± 2.4 | 80.0 ± 2.3 | 42.4 ± 2.5 |
| | DBGL | **84.3 ± 0.3** | **48.2 ± 1.3** | **83.4 ± 0.6** | **45.7 ± 1.2** | **83.4 ± 0.7** | **45.2 ± 0.7** | **82.3 ± 0.8** | **43.9 ± 1.5** | **81.1 ± 0.9** | **42.9 ± 1.0** |
| P19 | GRU-D | 88.5 ± 2.3 | 54.6 ± 3.7 | 88.8 ± 2.1 | 54.2 ± 3.4 | 88.0 ± 2.5 | 50.4 ± 7.5 | 87.5 ± 2.8 | 49.6 ± 6.9 | 86.4 ± 3.5 | 47.2 ± 8.6 |
| | mTAND | 79.6 ± 1.8 | 28.6 ± 1.9 | 79.2 ± 1.9 | 28.1 ± 2.1 | 78.0 ± 2.4 | 26.9 ± 2.9 | 77.2 ± 2.7 | 26.3 ± 2.9 | 76.2 ± 3.2 | 24.3 ± 4.8 |
| | DGM²-O | 87.4 ± 0.6 | 53.4 ± 1.5 | 87.3 ± 0.8 | 53.2 ± 1.7 | 86.6 ± 1.6 | 49.9 ± 5.1 | 85.8 ± 1.9 | 47.7 ± 5.9 | 85.2 ± 2.2 | 45.7 ± 6.7 |
| | MTGNN | 84.5 ± 1.4 | 48.9 ± 2.3 | 84.8 ± 1.7 | 49.8 ± 3.1 | 84.0 ± 1.9 | 47.2 ± 4.8 | 83.3 ± 2.2 | 45.5 ± 5.5 | 82.5 ± 2.9 | 42.7 ± 9.2 |
| | Raindrop | 88.2 ± 1.5 | 59.7 ± 1.5 | 88.1 ± 1.3 | 59.8 ± 1.4 | 87.8 ± 1.2 | 59.1 ± 1.7 | 87.6 ± 1.1 | 58.5 ± 1.9 | 87.1 ± 1.5 | 87.1 ± 1.5 |
| | DuETT | 85.2 ± 1.0 | 53.7 ± 1.0 | 84.8 ± 1.1 | 53.9 ± 0.8 | 84.7 ± 1.0 | 53.3 ± 1.6 | 84.3 ± 1.4 | 52.7 ± 2.1 | 84.4 ± 1.3 | 52.5 ± 2.0 |
| | Warpformer | 91.3 ± 0.8 | 55.2 ± 5.6 | 91.3 ± 0.8 | 55.1 ± 5.6 | 91.4 ± 0.8 | 56.0 ± 4.8 | 91.5 ± 0.7 | 56.4 ± 4.3 | 91.2 ± 0.8 | 56.2 ± 3.9 |
| | KEDGN | 91.3 ± 0.3 | 59.9 ± 0.7 | 91.2 ± 0.5 | 59.6 ± 0.9 | 90.9 ± 0.9 | 59.1 ± 1.1 | 90.7 ± 1.0 | 58.9 ± 1.2 | 90.1 ± 1.6 | 58.1 ± 2.0 |
| | DBGL | **92.8 ± 0.8** | **64.4 ± 1.2** | **91.5 ± 0.8** | **61.8 ± 1.4** | **91.4 ± 0.5** | **60.8 ± 2.6** | **91.2 ± 0.4** | **59.2 ± 1.8** | **90.4 ± 0.3** | **58.3 ± 1.8** |
| Physionet | GRU-D | 70.0 ± 3.0 | 32.1 ± 4.1 | 69.5 ± 2.6 | 31.1 ± 3.6 | 69.2 ± 3.0 | 31.0 ± 4.4 | 68.3 ± 3.6 | 30.1 ± 5.3 | 68.1 ± 3.7 | 29.8 ± 5.3 |
| | mTAND | 80.5 ± 2.1 | 42.8 ± 4.0 | 78.2 ± 3.4 | 40.5 ± 4.7 | 76.3 ± 4.0 | 37.7 ± 5.7 | 75.6 ± 3.9 | 36.6 ± 5.6 | 75.1 ± 3.9 | 36.1 ± 5.1 |
| | DGM²-O | 80.2 ± 0.9 | 38.6 ± 2.8 | 80.4 ± 0.9 | 38.3 ± 2.8 | 78.9 ± 1.9 | 37.1 ± 3.4 | 77.5 ± 3.7 | 35.4 ± 4.4 | 75.6 ± 5.0 | 34.0 ± 4.8 |
| | MTGNN | 68.9 ± 4.1 | 25.8 ± 4.8 | 69.3 ± 4.3 | 26.6 ± 4.5 | 69.0 ± 4.8 | 26.3 ± 5.2 | 68.3 ± 5.2 | 25.4 ± 4.8 | 67.2 ± 5.4 | 24.4 ± 4.8 |
| | Raindrop | 76.5 ± 1.2 | 33.4 ± 2.2 | 76.5 ± 1.3 | 32.3 ± 2.3 | 75.6 ± 2.0 | 30.8 ± 3.2 | 74.7 ± 2.6 | 29.7 ± 3.5 | 73.6 ± 3.2 | 28.8 ± 3.9 |
| | DuETT | 78.2 ± 2.8 | 39.9 ± 3.5 | 78.3 ± 3.0 | 39.9 ± 3.7 | 76.7 ± 3.7 | 37.9 ± 4.5 | 75.9 ± 3.8 | 37.0 ± 4.6 | 74.9 ± 4.3 | 35.9 ± 5.0 |
| | Warpformer | 78.2 ± 1.0 | 33.3 ± 2.1 | 77.7 ± 1.6 | 33.6 ± 1.8 | 75.8 ± 3.4 | 31.8 ± 3.0 | 73.8 ± 4.6 | 30.2 ± 4.1 | 72.7 ± 4.9 | 29.2 ± 4.2 |
| | KEDGN | 83.8 ± 1.0 | 49.4 ± 2.5 | 82.9 ± 2.5 | 48.0 ± 5.3 | 81.7 ± 2.8 | 46.4 ± 5.5 | 81.4 ± 2.5 | 45.8 ± 5.2 | 81.1 ± 2.4 | 45.2 ± 5.3 |
| | DBGL | **88.6 ± 0.9** | **57.8 ± 1.7** | **86.6 ± 1.5** | **53.6 ± 3.5** | **84.0 ± 2.2** | **50.1 ± 2.0** | **82.4 ± 1.9** | **48.1 ± 2.4** | **82.8 ± 1.5** | **46.7 ± 2.9** |

Table 21: Soft codebook utilization rates across different datasets.

| Dataset | P19 | Physionet | MIMIC-III | P12 |
|---|---|---|---|---|
| Utilization Rate | 0.4973 | 0.4977 | 0.4982 | 0.4978 |

These results confirm that the codebook is not merely a design embellishment but a robust mechanism that facilitates selective cross-sample communication, enhances learning of variable dependencies, and improves predictive performance when similarity structure is present.

## H  VISUALIZATION

Since DBGL builds a patient–variable bipartite graph, the learned variable nodes are expected to capture meaningful feature representations. To examine this, we visualize the variable embeddings using T-SNE (Maaten & Hinton (2008)). Following Luo et al. (2024), we group variables with similar temporal patterns and project their embeddings into a 2D space. The results, shown in Figure 6, reveal clear clustering that aligns with clinical groupings, suggesting that DBGL effectively encodes variable-specific information. As the PhysioNet dataset is a subset of P12 and shares the same variables, we report the visualization only on P12.

Here, the temporal patterns include sampling rate, sampling time, observation span, sequence length, and trends. Variables sharing similar temporal patterns are grouped and visualized with the same color. We observe that, after training, embeddings of variables within the same group are tightly clustered, indicating that DBGL effectively captures the temporal dynamics of irregular clinical variables. This ability to model variable-specific temporal patterns contributes to more informative representations and improved classification performance.

## I  EXPERIMENTS ON PAM DATASETS

To further evaluate the generalization of our proposed DBGL framework beyond binary clinical tasks, we conducted experiments on the PAM dataset (Reiss & Stricker (2012)), which contains 18 physical activities across 9 subjects, with 17 variables (IMU + heart rate) sampled over 600 time steps, with 60% missing ratio.

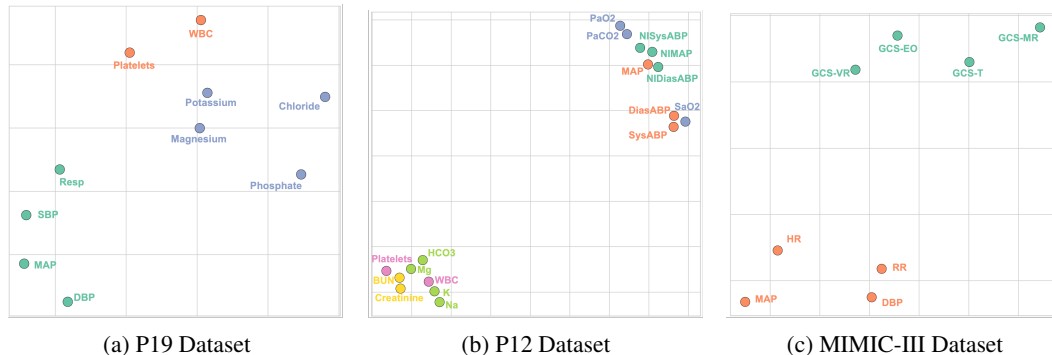

(a) P19 Dataset         (b) P12 Dataset         (c) MIMIC-III Dataset

Figure 6: T-SNE visualization of partial variable representations on three datasets.

Table 22: Performance comparison of different models on the dataset. Values are reported as mean $\pm$ std (%).

| Model | Accuracy (%) | Precision (%) | Recall (%) | F1 score (%) |
|---|---|---|---|---|
| Transformer | $83.5 \pm 1.5$ | $84.8 \pm 1.5$ | $86.0 \pm 1.2$ | $85.0 \pm 1.3$ |
| Trans-mean | $83.7 \pm 2.3$ | $84.9 \pm 2.6$ | $86.4 \pm 2.1$ | $86.4 \pm 2.1$ |
| GRU-D | $83.3 \pm 1.6$ | $84.6 \pm 1.2$ | $84.6 \pm 1.2$ | $84.8 \pm 1.2$ |
| SeFT | $67.1 \pm 2.2$ | $70.0 \pm 2.4$ | $68.2 \pm 1.5$ | $68.5 \pm 1.8$ |
| mTAND | $74.6 \pm 4.3$ | $74.3 \pm 4.0$ | $79.5 \pm 2.8$ | $76.8 \pm 3.4$ |
| IP-Net | $74.3 \pm 3.8$ | $75.6 \pm 2.1$ | $77.9 \pm 2.2$ | $76.6 \pm 2.8$ |
| DGM$^2$-O | $82.4 \pm 2.3$ | $85.2 \pm 1.2$ | $83.9 \pm 2.3$ | $84.3 \pm 1.8$ |
| MTGNN | $83.4 \pm 1.9$ | $85.2 \pm 1.7$ | $86.1 \pm 1.9$ | $85.9 \pm 2.4$ |
| RainDrop | $88.5 \pm 1.5$ | $89.9 \pm 1.5$ | $89.9 \pm 0.6$ | $89.8 \pm 1.0$ |
| ViTST | $95.8 \pm 1.3$ | $96.2 \pm 1.3$ | $96.2 \pm 1.3$ | $96.5 \pm 1.2$ |
| TimeCHEAT | $96.5 \pm 0.6$ | $97.1 \pm 0.5$ | $96.9 \pm 0.6$ | $97.0 \pm 0.5$ |
| DBGL | $90.4 \pm 0.9$ | $92.5 \pm 0.7$ | $92.4 \pm 0.9$ | $92.4 \pm 0.8$ |

We applied DBGL for classification. Unlike clinical datasets, PAM variables are dense and regularly sampled, with minimal variable-specific decay irregularity. Applying the decay-aware module here would introduce noise and degrade performance.

DBGL achieves strong performance (92.4 F1), surpassing most irregular-time baselines. Its decay-aware module provides limited benefit here due to the lack of variable-wise decay heterogeneity, whereas ViTST and TimeCHEAT exploit strong generic sequence modeling (patching, global Transformers) suitable for dense continuous signals.

Our decay-rate analysis confirms this: almost all variables exhibit extremely small decay values ($\lambda \approx 0.006 \sim 0.01$ ), indicating negligible heterogeneity. This aligns with the characteristics of PAM as dense motion sensor data without irregular sampling.

These results indicate that DBGL can generalize to diverse time series, even when the decay heterogeneity assumption does not hold.

## J    THE USAGE OF LARGE LANGUAGE MODELS LLM

We employed large language models (LLMs) in a limited capacity, strictly for writing assistance tasks such as grammar correction, style refinement, and table formatting. The LLM was also used solely to generate code comments and explanatory notes in the supplementary implementation, and no core experimental code was produced by the LLM. All main model implementations, data pro-

Table 23: Variable-specific decay rates ($\lambda$) for each variable.

| Variable | $\lambda$ | Variable | $\lambda$ |
|---|---|---|---|
| v0 | 1.0451 | v1 | 0.00887 |
| v2 | 0.00662 | v3 | 0.05265 |
| v4 | 0.00910 | v5 | 0.00660 |
| v6 | 0.04641 | v7 | 0.04618 |
| v8 | 0.00823 | v9 | 0.00720 |
| v10 | 0.00836 | v11 | 0.02725 |
| v12 | 0.01110 | v13 | 0.00728 |
| v14 | 0.00806 | v15 | 0.00756 |
| v16 | 0.00714 | — | — |

cessing pipelines, and training routines were entirely written and verified by the authors. All suggested edits were carefully reviewed and selectively incorporated by the authors. The scientific content, ideas, analyses, and conclusions presented in this paper are entirely our own. The authors take full responsibility for the paper's content, including any remaining errors or inaccuracies.

