# OpenReview forum: "DBGL: Decay-aware Bipartite Graph Learning for Irregular Medical Time Series"
_ICLR.cc/2026/Conference — Submitted to ICLR 2026_

### Official Review · Reviewer_nFwr · 2025-10-26

**Soundness:** 2
**Presentation:** 3
**Contribution:** 2
**Rating:** 2
**Confidence:** 4

**Summary:**

The paper introduces DBGL to solve “temporal sampling irregularity” using a bipartite graph and “variable decay irregularity” using node-specific temporal decay encoding. Experiments demonstrate competitive performance over state-of-the-art models.

**Strengths:**

1.	Code is provided in supplementary materials and is runnable.

2.	Baselines included in experiments are up to date.

**Weaknesses:**

1.	The paper’s motivation is clearly problematic. First of all, “distort temporal sampling irregularity” at line 15 is already solved by the bipartite graph representation proposed in GraFITi[1] and adopted by TimeCHEAT[2] (included in baselines). DBGL merely changes the existing time-variable bipartite graph to a variable-patient one. Secondly, “variable decay irregularity” at line 16 is a problem specific to RNN-based models and is not a characteristic of irregular time series. The concept of “decay” is related to the forgetting mechanism in GRU-D, so the claim “most methods impose uniform or overly simplistic decay assumptions” at line 65 does not apply to methods like graph-based (TimeCHEAT[2], Raindrop[5]) or attention-based ones (SeFT[4]).

2.	Some claims in the paper are biased or wrong. (1) The claim “two fundamental limitations persist in existing approaches. One is the distortion of inherent irregularity, where resampling eliminates informative missingness…” at line 62 assumes resampling is a problem cannot be solved by existing methods, while TimeCHEAT[2], GRU-D[3], SeFT[4], Raindrop[5] did not use resampling at all. (2) The claim “variable-variable modeling neglects asynchronous interactions between patients and variables…” at line 63 is confusing. Since “patients” are essentially equal to “samples” in medical irregular time series datasets, this phrase is equal to “variable-variable modeling neglects sample-variable modeling”. They are two independent types of modelling perspectives and are not related at all. Therefore, the logic of this claim is chaotic.

3.	DBGL is composed of two main components: a patient-variable bipartite graph and a modified version of RNN. Patient-variable bipartite graph seems to be responsible for variable dependency learning, while RNN is responsible for temporal one. However, since the time-variable bipartite graph in GraFITi[1] and TimeCHEAT[2] can already model temporal and variable dependencies simultaneously without any problems in “temporal sampling irregularity” or “variable decay irregularity” (see explanations in weakness 1), DBGL’s more complicated design seems to be inferior.

4.	Patient-variable bipartite graph in Figure 1 does not provide any useful information in illustrating how it works. Graphs from t1 to tn look exactly the same. If the plotted IMTS input on the top left is p1, then why are p1 and v2 connected in the bipartite graph at t2? The green v2 does not have observation at t2.

5.	DBGL runs extremely slow using the codes provided in supplementary materials, similar to the speed of ODE-based models, which raises doubts about whether the design mechanisms are worth the trade-off.

6.	Implementation codes in supplementary materials seem to include lines written by LLMs, while the disclaim in Appendix G does not mentioned its usage.

[1] V. K. Yalavarthi et al.; “GraFITi: Graphs for Forecasting Irregularly Sampled Time Series”; AAAI 2024

[2] J. Liu, M. Cao, and S. Chen; “TimeCHEAT: A Channel Harmony Strategy for Irregularly Sampled Multivariate Time Series Analysis”; AAAI 2025

[3] Z. Che, S. Purushotham, K. Cho, D. Sontag, and Y. Liu; “Recurrent Neural Networks for Multivariate Time Series with Missing Values”; Sci Rep 2018

[4] M. Horn, M. Moor, C. Bock, B. Rieck, and K. Borgwardt; “Set Functions for Time Series”; ICML 2020

[5] X. Zhang, M. Zeman, T. Tsiligkaridis, and M. Zitnik; “Graph-Guided Network for Irregularly Sampled Multivariate Time Series”; ICLR 2022

**Questions:**

1.	What’s the point of using RNN in DBGL? As mentioned in weakness 3, RNN appears redundant compared to the time-variable bipartite graph design in GraFITi and TimeCHEAT. The bipartite graph natively handles irregularities, while RNNs require carefully designed decay functions to compensate for them.

2.	To use patient-variable bipartite graph for learning variable dependencies across samples, a key challenge is to ensure that messages passed from other samples can promote the learning in current sample, which require samples to be similar enough or belong to the same category. Does DBGL have any related design?

**Details Of Ethics Concerns:**

None.

---

> ### Author Response · Authors · 2025-11-18
> **Response to Reviewer nFwr (part 1/4)**
>
> **Q1:** About the motivation of our DBGL
>
> **A1:** We thank the reviewer for the detailed feedback. We agree that GraFITi and TimeCHEAT adopt bipartite graph representations to handle temporal sampling irregularity. However, our approach goes beyond prior works in several important aspects:
>
> **1. Patient–Variable Bipartite Graph and Temporal Sequence:**
> Instead of a standard variable–time bipartite graph, we construct a patient–variable bipartite graph and maintain a time-varying sequence of graphs. This design explicitly captures cross-subject heterogeneity in variable behaviors and allows temporal propagation that respects both individual patient patterns and variable-specific dynamics, which traditional variable–time graphs cannot directly model.
>
>
> **2. Decay Rate λ Definition and Variable-Specific Temporal Decay:**
> Decay Rate λ for each clinical variable: where a higher λ indicates a fast-changing variable (e.g., HR, BP) and a lower λ corresponds to a more stable variable (e.g., creatinine, hemoglobin). This metric provides a unified quantitative measure of clinically meaningful differences in temporal behavior across variables.
>
> Motivated by this heterogeneity, our variable-specific temporal decay encoding assigns time-sensitive update weights to each graph node according to its corresponding λ. While decay mechanisms are often associated with GRU-D, our approach is independent of RNN forget gates and directly leverages the empirical variable-specific temporal dynamics. By explicitly modeling these decays, DBGL can adaptively weigh historical states, improving representation under long intervals or sparse observations—something prior graph- or attention-based methods like TimeCHEAT or SeFT do not capture.
>
> **3.Empirical Evidence of Heterogeneous Decay:**
> To directly assess whether our model captures clinically meaningful temporal dynamics, we conducted a systematic analysis of variable-specific decay rates (λ_{p,n}^t) across all datasets.
>
> **Representative Results (P12 dataset):**
>
> | Variable | λ  | Variable | λ  | Variable | λ  |
> |----------|----|---------|----|---------|----|
> | HR       | 1.73 | Temp     | 0.0265 | Creatinine | 0.1399 |
> | Lactate  | 13.00 | ALP     | 14.71 | PaCO2   | 0.0225 |
> | ALT      | 14.62 | AST    | 13.89 | MAP     | 0.0913 |
>
> Kruskal–Wallis test confirms significant heterogeneity across variables (p = 3.87 × 10^{-118}), indicating that variables exhibit distinct temporal decay patterns. Fast-changing vital signs show high λ, while stable lab tests exhibit low λ, aligning with clinical intuition. Some lab tests (e.g., ALT, AST, lactate) can change rapidly under acute conditions, highlighting the necessity of variable-specific decay modeling.
>
> **Other datasets:** Similar patterns are observed in P19, Physionet, and MIMIC-III (all p ≪ 0.001). To maintain brevity in this rebuttal, we present only representative examples; the full analysis across all variables and datasets will be included in the revised manuscript.
>
> **Conclusion:** These analyses validate that our model learns variable-specific decay patterns that are clinically meaningful and heterogeneous. Uniform decay assumptions would fail to capture these differences, whereas our variable-specific decay encoding module enables more accurate representation of clinical time series and improved downstream performance.
>
> **4.Motivation Validity:**
> These analyses justify the need for variable-specific decay modeling. Using a uniform decay assumption would fail to capture heterogeneous behaviors, treating stable and reactive biomarkers equally. By learning decay rates directly from data, DBGL adaptively captures these irregular dynamics, improving temporal representation and downstream predictive performance.
>
> **5.Summary:**
> DBGL is not a mere reuse of bipartite graphs. By combining patient–variable graphs with variable-specific decay encoding, it models both cross-subject heterogeneity and variable-specific temporal dynamics, addressing challenges not captured by prior graph- or attention-based methods.
>
>
> **Q2:** Some claims in the paper are biased or wrong.
>
> **A2:** We thank the reviewer for the feedback. We acknowledge that the original statements were imprecise: methods such as TimeCHEAT, GRU-D, SeFT, and Raindrop do not rely on resampling. Our intention was to highlight that handling irregular observation intervals and missing values remains challenging, and potential information loss can occur in sequence-flattening approaches or naive imputation.
>
> Our patient–variable bipartite graph explicitly introduces patient nodes, enabling the model to capture cross-patient differences for the same variable. This allows personalized modeling of variable dynamics across individuals, which is not possible with standard variable–variable or variable–time graphs. In other words, patient nodes serve as aggregation units, enhancing the model’s ability to learn heterogeneous patterns across patients and variables.

---

> ### Author Response · Authors · 2025-11-18
> **Response to Reviewer nFwr (part 2/4)**
>
> **Continue to A2:**
> From this perspective, our approach represents a modeling extension: patient nodes serve as units for information aggregation and propagation, enhancing the ability to model dynamic heterogeneity across patients and variables.
>
> Based on this clarification, we have revised the logic and phrasing of this section in the manuscript accordingly.
>
> Irregular sampling is a central challenge in multivariate clinical time-series. Traditional interpolation or resampling pipelines produce regularly spaced sequences but may obscure informative missingness patterns. Recent transformer- and diffusion-based approaches incorporate continuous-time embeddings or learned interpolation to mitigate these issues, yet they still operate primarily on flattened sequences and therefore struggle to represent which variable is observed at which time, a key structural signal in clinical data. Bipartite variable–time graphs make this structure more explicit by linking variable nodes to time nodes, but they are typically constructed within local windows and do not capture patient-specific sampling behaviors. Moreover, clinical variables naturally exhibit heterogeneous temporal dynamics, some change within minutes, others over hours or days, yet most existing models encode time uniformly and lack explicit mechanisms for variable-specific decay.
>
> In summary, two main limitations remain in existing approaches. First, preprocessing or flattened sequence modeling can obscure inherent irregularity: resampling may eliminate informative missingness, and sequence-based variable modeling captures limited asynchronous interactions between patients and variables. Second, temporal dynamics are often treated uniformly across variables, ignoring that clinical variables evolve at heterogeneous rates and exhibit variable-specific sensitivity to elapsed time. This simplification can lead to incomplete or less informative patient representations for downstream clinical tasks.
>
> **Q3:** Main components of DBGL.
>
> **A3:** In GraFITi [1] and TimeCHEAT [2], the time–variable bipartite graph updates each time node by aggregating all observed variables at that time, and each variable node by aggregating all time points where it is observed. While this structure captures variable–time interactions, it cannot explicitly model **variable-specific temporal decay**: clinical variables evolve on different time scales (e.g., heart rate changes every minute, creatinine over hours or days). Consequently, dependencies across consecutive time points are not dynamically encoded, limiting the model’s ability to represent fast- vs. slow-changing physiological signals.
>
>
> Modeling such temporal dependencies is critical in clinical scenarios, as it directly affects the model’s ability to capture the persistence and rate of change of patient states, which in turn influences the reliability of downstream clinical predictions.
>
> To address this, our DBGL shifts the modeling perspective from variable–time to patient–variable, constructing a sequence of patient–variable bipartite graphs. At each time step:
>
> 1. Patient nodes connect to currently observed variable nodes and aggregate multi-variable information via graph message passing.
>
> 2. Over time, the patient node states are dynamically updated via decay functions and recurrent units, explicitly capturing temporal continuity and asynchronous variable changes.
>
> This design offers several advantages:
>
> 1. Dynamic dependency modeling: patient nodes share hidden states across consecutive time steps, enabling explicit modeling of temporal evolution rather than relying solely on sparse time-node aggregation.
>
> 2. Variable-specific decay rate adaptation: the decay mechanism combined with patient node memory allows fast-changing variables (e.g., vital signs) and slow-changing variables (e.g., lab measurements) to update at different rates within the unified graph, reflecting real physiological dynamics.
>
> 3. Cross-variable information integration: variable nodes are shared across patients, allowing the model to learn statistical dependencies at the population level, while patient nodes capture individual-specific patterns.
>
> We further compared DBGL with TimeCHEAT on classification performance (AUROC/AUPRC):
> | Method      | P19 AUROC | P19 AUPRC | P12 AUROC | P12 AUPRC |
> |------------|-----------|-----------|-----------|-----------|
> | TimeCHEAT  | 89.5 ± 1.9 | 56.1 ± 4.6 | 84.5 ± 0.7 | 48.2 ± 1.9 |
> | DBGL       | 93.3 ± 0.5 | 66.3 ± 1.4 | 88.1 ± 0.4 | 56.3 ± 1.0 |
> | Gains      | +3.8       | +10.2      | +3.6       | +8.1       |
>
> As shown, our approach achieves substantial performance gains, demonstrating that DBGL’s more sophisticated design effectively captures temporal continuity and asynchronous variation patterns, while leveraging cross-patient information for more accurate modeling of irregular clinical time series.

---

> ### Author Response · Authors · 2025-11-18
> **Response to Reviewer nFwr (part 3/4)**
>
> **Q4:** About Figure 1.
>
> **A4:** We thank the reviewer for the valuable suggestion. In the revised manuscript, Figure 1 has been corrected and improved. Our original figure did not clearly illustrate the dynamic construction of the patient–variable bipartite graph sequence, which may have caused confusion regarding node connections over time.
>
> In the updated figure:
>
> 1. The IMTS input in the top-left corner is now explicitly aligned with the corresponding graph sequence on the right.
>
> 2. Only variables that are observed at a given time step are connected to the patient node, ensuring that connections such as p1–v2 at t2 only appear if v2 is actually measured at t2.
>
> 3. For the construction details of the edges in the patient-variables graph of more patient 1 cases, we have distinguished the edge variables with different colors to better facilitate understanding.
>
> These modifications clarify the graph construction process and demonstrate how patient–variable interactions are dynamically captured, addressing the reviewer’s concerns while better illustrating the underlying methodology.
>
> **Q5:** DBGL runs extremely slow.
>
> **A5:** We provide a comparison of training time and memory usage in Table 6 of the appendix. Our method incurs only a slight increase in training time and memory compared to KEDGN, while achieving substantial performance gains. Therefore, we believe the additional computational cost introduced by our design is justified. It is also worth noting that KEDGN achieves its reported performance by leveraging an additional large language model (LLM), whose computational cost is not included in our comparison.
>
> Futhermore, we conduct bottleneck analysis on a single 24GB NVIDIA GeForce RTX 4090 GPU using the P12 dataset, exploring the impact of batch size (B), codebook size (C), sequence length (L), and number of variables (V). To vary L and V, we repeated sequences and variables accordingly to increase sequence length and variable count. We report both GPU memory consumption and training time per epoch. Our baseline configuration is B=256, V=36, L=215, C=4096, and when varying one parameter, the others are kept constant.
>
> | B    | Space (MiB) | Time (s/epoch) | V    | Space (MiB) | Time (s/epoch) | L    | Space (MiB) | Time (s/epoch) | C    | Space (MiB) | Time (s/epoch) |
> |-|-|-|-|-|-|-|-|-|-|-|-|
> | 128  | 2670        | 56             | 36   | 3806        | 30             | 215  | 3806        | 30             | 512  | 2950        | 28             |
> | 256  | 3806        | 30             | 72   | 5258        | 35             | 430  | 6294        | 66             | 1024 | 3064        | 28             |
> | 512  | 6034        | 18             | 144  | 8154        | 48             | 860  | 11254       | 192            | 2048 | 3238        | 29             |
> | 1024 | 10776       | 12             | 288  | 14208       | 71             | 1290 | 16316       | 338            | 4096 | 3806        | 30             |
> | 2048 | 19700       | 9              | 432  | 20306       | 89             | 1720 | 21288       | 525            | 8172 | 4968        | 31             |
>
> The results show that increasing batch size B significantly increases GPU memory usage (e.g., from 2670 MiB at B=128 to 19700 MiB at B=2048), while training time per epoch decreases due to higher GPU utilization at larger batch sizes. Increasing the number of variables V also substantially increases memory usage (from 3806 MiB at V=36 to 20306 MiB at V=432) and training time (from 30 s to 89 s per epoch). The sequence length L has the most pronounced effect on both memory and training time; for example, increasing L from 215 to 1720 raises memory from 3806 MiB to 21288 MiB and epoch time from 30 s to 525 s. In contrast, codebook size C has a relatively minor impact on memory and training time (e.g., increasing C from 512 to 8172 only increases memory from 2950 MiB to 4968 MiB, with negligible change in epoch time).
>
> Finally, we also conduct the training time and testing time comparison between DBGL and KEDGN using the same GPU (a NVIDIA A800 80GB PCIe). The measured testing time (in seconds) is summarized below:
>
> | Training Time (s/epoch) | P19 | Physionet | MIMIC-III | P12 |
> |-|-|-|-|-|
> | KEDGN                  | 6.6 | 8.4       | 36.6      | 17.4 |
> | DBGL                   | 10.8 | 9.6       | 50.4      | 30.0 |
>
> | Inference Time (s/epoch) | P19 | Physionet | MIMIC-III | P12 |
> |-|-|-|-|-|
> | KEDGN | 1.76 | 0.93 | 5.92 | 1.61 |
> | DBGL  | 2.03 | 0.81 | 4.86 | 1.92 |
> Several observations can be drawn from the results:
>
> The inference time of DBGL is highly comparable to that of the sequence-based KEDGN.
> On P19 and P12, DBGL is slightly slower (+0.27s and +0.31s), reflecting the lightweight online graph construction.
> On Physionet and MIMIC-III, DBGL is even faster (−0.12s and −1.06s), showing that dynamic graph generation does not introduce a significant computational burden.

---

> ### Author Response · Authors · 2025-11-18
> **Response to Reviewer nFwr (part 4/4)**
>
> **Continue to A5**:
> The dynamic graph building process adds only minor overhead because: graph nodes correspond directly to observed variables at each step, and the graph size remains small (patient node + variable nodes),
> resulting in linear complexity with respect to observed variables.
>
> Across all datasets, the runtime difference between DBGL and sequence baselines is within 0.3–1.0 seconds, demonstrating that the proposed graph mechanism is both practical and efficient for real-world healthcare scenarios.
>
> **Q6:** The usage of LLM in code.
>
> **A6:** We thank the reviewer for pointing this out. For clarification, the LLM was used solely to generate code comments and explanatory notes in the supplementary implementation, and no core experimental code was produced by the LLM. All main model implementations, data processing pipelines, and training routines were entirely written and verified by the authors. We apologize for the omission in Appendix G and will update the disclaimer in the next version to accurately reflect this usage.
>
> **Q7:** What’s the point of using RNN in DBGL?
>
> **A7:** DBGL is designed to address two complementary types of temporal irregularity in clinical time series: sampling irregularity and variable-specific decay irregularity.
>
> The patient–variable bipartite graph effectively handles sampling irregularity by explicitly modeling which variables are observed at each time step. However, it cannot capture how different variables evolve over time—that is, their distinct temporal autocorrelation or decay behaviors. For example, heart rate changes rapidly, whereas creatinine evolves slowly. We refer to this phenomenon as decay irregularity.
>
> To model decay irregularity, we incorporate an RNN-based temporal encoder, which enables the model to:
> 1. Embed variable-specific decay rates into the temporal feature extraction, rather than treating decay as an external correction term.
>
> 2. Transform decay irregularities into informative temporal features, instead of noise that must be compensated for.
>
> 3. Leverage long-range temporal dependencies, which complement the structural information encoded by the bipartite graph.
>
> In short, the bipartite graph and RNN play distinct but complementary roles. The graph captures when each measurement occurs, while the RNN captures how each variable evolves over time. This combination allows DBGL to generate clinically meaningful representations and improves predictive performance, particularly for variables with heterogeneous temporal dynamics.
>
> **Q8:** Related design about similar samples learning.
>
> **A8:** DBGL includes a dedicated mechanism to ensure that cross-sample information is selectively transferred only from samples that exhibit similar variable behaviors. This mechanism is realized through our learned codebook, which plays a crucial role in regulating message passing on the patient–variable bipartite graph.
>
> Specifically, the codebook serves two complementary purposes:
>
> (1) Soft codebook fusion for shared variable semantics
>
> Each variable embedding is softly projected onto a set of learned code vectors.
> This soft utilization allows variables with similar clinical semantics—such as laboratory tests with similar dynamic patterns—to map to similar code entries.
> As a result:
>
> Samples with similar variable states naturally obtain closer code representations.
>
> Only semantically aligned information is aggregated in the bipartite graph.
>
> Thus, even without explicit clustering labels, DBGL forms latent variable-level neighborhoods that guide reliable and clinically meaningful message passing.
>
> (2) Nearest-code matching for cross-sample alignment
>
> During message passing, DBGL further uses the nearest code vector to anchor each variable embedding.
> This ensures that:
>
> Messages are aggregated primarily from samples whose variables lie near the same code prototype.
>
> Information from dissimilar samples is naturally suppressed due to low codebook similarity.
>
> This design effectively avoids the risk of “negative transfer” from heterogeneous samples.

---

> > ### Comment · Reviewer_nFwr · 2025-11-22
> >
> > Thank the authors for their detailed response, which addressed some of my concerns, especially clarifications regarding the motivation. After considering the other reviewers’ comments, I encourage the authors to thoroughly revise the manuscript based on all feedback before submitting it to another conference or journal. Therefore, I will increase my overall score from 2 to 4, and wish the authors the best of luck in their next submission.

---

> > > ### Author Response · Authors · 2025-11-22
> > > **Response to Reviewer nFwr**
> > >
> > > Thank you very much for your positive response.
> > > Regarding the comments raised by the other reviewers, we have conducted targeted analyses, clarifications, and revisions—while keeping our core methodology unchanged—and we have provided detailed responses as well as additional experiments in the updated version. We believe these revisions have thoroughly and reasonably addressed concerns raised during the review process, including those that may also be related to your points of interest.
> > >
> > > If you have any remaining questions or concerns regarding our responses or revisions, we would be very happy to further communicate and discuss them.
> > > If, after reviewing the updated manuscript, you find that the overall quality of the paper has been improved, we would greatly appreciate any reconsideration of your evaluation as you see appropriate.
> > >
> > > We sincerely look forward to any additional feedback you may have, as it would be highly valuable for further improving our work.

---

### Official Review · Reviewer_qRN1 · 2025-10-29

**Soundness:** 2
**Presentation:** 2
**Contribution:** 2
**Rating:** 6
**Confidence:** 4

**Summary:**

This paper addresses the problem of modeling irregularly sampled clinical time series. The authors propose constructing a Patient–Variable Bipartite Graph (PVG) at each time step, where edges correspond to actual observations, and using an autoregressive update to maintain variable-specific hidden states over time. The model further introduces a Temporal Decay Encoding (TDE) mechanism to capture variable-specific temporal dynamics and a learnable codebook to regularize patient representations.

**Strengths:**

1. The idea of jointly modeling patients and variables via a bipartite graph is reasonable. It provides a structured way to represent irregular observations and aligns well with clinical intuition.

2. The writing and organization are clear, with a well-structured.

3. The study offers extensive experiments.

**Weaknesses:**

1. The model builds on key hypotheses, e.g., that patient–variable relations better capture irregularity and that each variable should have its own decay rate, but these remain speculative. Providing more concrete evidence will make these arguments more convincing.

 2. The TDE employs an exponential decay function with a learned Softplus(MLP) rate. However, there is no discussion or comparison with alternative continuous-time kernels or analysis of identifiability and numerical stability under large time intervals.

3. The evaluation primarily targets mortality prediction on small-scale clinical datasets (<40 variables). This restricts generalizability. Testing on more diverse or large-scale tasks (e.g., phenotype classification, readmission) would better demonstrate practical applicability.

4. Constructing a patient–variable bipartite graph at each timestep and running EdgeSAGE message passing is computationally expensive. The experiments use relatively few variables, and the paper does not report graph construction overhead.

5. The paper suggests that most Transformer-based methods rely on resampling or imputation, which is misleading. Models such as SeFT, StraTS, and Warpformer also handle irregular sampling directly. The related work section should be more comprehensive and precise.

**Questions:**

1. During inference, are the patient–variable graphs pre-built or dynamically generated? If generated online, what is the computational overhead and runtime complexity compared to sequence-based baselines?

2. The matched code vector appears to contribute marginally based on ablation results. Is the codebook essential to model performance or mainly a design embellishment? Could the authors provide additional analysis (e.g., utilization rate, collapse behavior) to support its utility?

3. Could the authors provide quantitative evidence isolating the contributions of Time Embedding versus Temporal Decay Encoding? Their roles appear conceptually overlapping.

---

> ### Author Response · Authors · 2025-11-18
> **Response to Reviewer qRN1 (part 1/5)**
>
> **Q1**: Key hypotheses verification.
>
> **A1**:  Thank you for raising this point. To directly assess whether our model captures clinically meaningful temporal dynamics, we conducted a systematic analysis of variable-specific decay rates (λ_{p,n}^t) across all datasets.
>
> **Decay Rate λ Definition:** Higher λ indicates fast-changing variables (e.g., HR, BP), while lower λ corresponds to more stable variables (e.g., creatinine, hemoglobin). This metric therefore provides a unified quantitative way to capture clinically meaningful differences in temporal behavior across variables.
>
> **Representative Results (P12 dataset):**
>
> | Variable | λ  | Variable | λ  | Variable | λ  |
> |----------|----|---------|----|---------|----|
> | HR       | 1.73 | Temp     | 0.0265 | Creatinine | 0.1399 |
> | Lactate  | 13.00 | ALP     | 14.71 | PaCO2   | 0.0225 |
> | ALT      | 14.62 | AST    | 13.89 | MAP     | 0.0913 |
>
> Kruskal–Wallis test confirms significant heterogeneity across variables (p = 3.87 × 10^{-118}), indicating that variables exhibit distinct temporal decay patterns. Fast-changing vital signs show high λ, while stable lab tests exhibit low λ, aligning with clinical intuition. Some lab tests (e.g., ALT, AST, lactate) can change rapidly under acute conditions, highlighting the necessity of variable-specific decay modeling.
>
> **Other datasets:** Similar patterns are observed in P19, Physionet, and MIMIC-III (all p ≪ 0.001). To maintain brevity in this rebuttal, we present only representative examples; the full analysis across all variables and datasets will be included in the revised manuscript.
>
> **Conclusion:** These analyses validate that our model learns variable-specific decay patterns that are clinically meaningful and heterogeneous. Uniform decay assumptions would fail to capture these differences, whereas our variable-specific decay encoding module enables more accurate representation of clinical time series and improved downstream performance.
>
> **Q2:** Kernel funcition ablation in TDE
>
> **A2:** Thank you for the comment. Indeed, the TDE employs an exponential decay function with a learned Softplus(MLP) rate. To investigate the effect of alternative continuous-time kernels, we conducted additional experiments by replacing the original exponential decay:
>
> **Gaussian Kernel**: $\gamma=exp(-(\lambda \cdot \Delta t)^{2})$,
>
> **Linear Kernel**: $\gamma = max(1-\lambda \cdot \Delta t, 0)$,
>
> while keeping the MLP-learned decay rate. We compared these with the standard MLP + exponential kernel and the pure exponential kernel without MLP. Averaged across all datasets, the results are:
> | Kernel        | P19 AUROC | P19 AUPRC | Physionet AUROC | Physionet AUPRC | MIMIC-III AUROC | MIMIC-III AUPRC | P12 AUROC | P12 AUPRC | Avg. AUROC | Avg. AUPRC |
> |---------------|-----------|-----------|----------------|----------------|----------------|----------------|------------|------------|-------------|-------------|
> | MLP + Linear   | 93.2 ± 0.7 | 65.6 ± 0.9 | 89.3 ± 0.4 | 60.8 ± 2.2 | 84.8 ± 0.2 | 49.5 ± 1.0 | 87.6 ± 0.3 | 54.2 ± 0.5 | 88.73 | 57.53 |
> | MLP + Gaussian | 93.3 ± 0.4 | 66.1 ± 1.6 | 89.3 ± 0.4 | 60.7 ± 2.4 | 84.6 ± 0.9 | 48.3 ± 2.5 | 87.7 ± 0.5 | 54.5 ± 1.3 | 88.73 | 57.40 |
> | MLP + Exp      | 93.3 ± 0.5 | 66.3 ± 1.4 | 89.1 ± 0.3 | 60.8 ± 2.2 | 85.2 ± 0.4 | 50.2 ± 1.0 | 88.1 ± 0.4 | 56.3 ± 1.0 | 88.93 | 58.40 |
> | Exp            | 92.9 ± 0.3 | 65.2 ± 0.6 | 88.8 ± 0.7 | 58.2 ± 1.9 | 84.9 ± 0.2 | 49.9 ± 0.5 | 87.0 ± 0.3 | 53.0 ± 1.0 | 88.40 | 56.58 |
>
> The MLP + exponential kernel achieves the best overall performance, confirming the benefit of learning variable-specific decay rates. Alternative kernels perform slightly worse, demonstrating DBGL’s robustness to kernel choice.
>
> **Numerical Stability & Identifiability:**
> Softplus ensures decay_rate ≥ 0, so $exp(-\lambda * Δt) ∈ (0,1]$, preventing overflow/underflow even for large Δt. γ decreases monotonically with Δt and varies with variable-specific edge attributes via the MLP, mapping variables with different temporal dynamics to distinct decay behaviors. This ensures both smooth gradients and identifiability of variable dynamics.
>
> In summary, the combination of Softplus and the exponential function guarantees a smooth, differentiable, and numerically stable decay, while effectively capturing heterogeneous temporal dynamics across variables.

---

> ### Author Response · Authors · 2025-11-18
> **Response to Reviewer qRN1 (part 2/5)**
>
> **Q3**: Testing on more diverse or large-scale tasks
>
> **A3**: To further evaluate the generalization of our proposed DBGL framework beyond binary clinical tasks, we conducted experiments on the PAM dataset [1], which contains 18 physical activities across 9 subjects, with 17 variables (IMU + heart rate) sampled over 600 time steps, with 60\% missing ratio.
>
> We applied DBGL for classification. Unlike clinical datasets, PAM variables are dense and regularly sampled, with minimal variable-specific decay irregularity. Applying the decay-aware module here would introduce noise and degrade performance.
>
> | Model        | Accuracy (%)      | Precision (%)     | Recall (%)        | F1 score (%)     |
> |--------------|-----------------|-----------------|-----------------|----------------|
> | Transformer  | 83.5 ± 1.5       | 84.8 ± 1.5       | 86.0 ± 1.2       | 85.0 ± 1.3     |
> | Trans-mean   | 83.7 ± 2.3       | 84.9 ± 2.6       | 86.4 ± 2.1       | 86.4 ± 2.1     |
> | GRU-D        | 83.3 ± 1.6       | 84.6 ± 1.2       | 84.6 ± 1.2       | 84.8 ± 1.2     |
> | SeFT         | 67.1 ± 2.2       | 70.0 ± 2.4       | 68.2 ± 1.5       | 68.5 ± 1.8     |
> | mTAND        | 74.6 ± 4.3       | 74.3 ± 4.0       | 79.5 ± 2.8       | 76.8 ± 3.4     |
> | IP-Net       | 74.3 ± 3.8       | 75.6 ± 2.1       | 77.9 ± 2.2       | 76.6 ± 2.8     |
> | DGM²-O       | 82.4 ± 2.3       | 85.2 ± 1.2       | 83.9 ± 2.3       | 84.3 ± 1.8     |
> | MTGNN        | 83.4 ± 1.9       | 85.2 ± 1.7       | 86.1 ± 1.9       | 85.9 ± 2.4     |
> | RainDrop     | 88.5 ± 1.5       | 89.9 ± 1.5       | 89.9 ± 0.6       | 89.8 ± 1.0     |
> | ViTST        | 95.8 ± 1.3       | 96.2 ± 1.3       | 96.2 ± 1.3       | 96.5 ± 1.2     |
> | TimeCHEAT    | 96.5 ± 0.6       | 97.1 ± 0.5       | 96.9 ± 0.6       | 97.0 ± 0.5     |
> | DBGL         | 90.4 ± 0.9       | 92.5 ± 0.7       | 92.4 ± 0.9       | 92.4 ± 0.8     |
>
>
> DBGL achieves strong performance (92.4 F1), surpassing most irregular-time baselines. Its decay-aware module provides limited benefit here due to the lack of variable-wise decay heterogeneity, whereas ViTST and TimeCHEAT exploit strong generic sequence modeling (patching, global Transformers) suitable for dense continuous signals.
>
>
> Our decay-rate analysis confirms this: almost all variables exhibit extremely small decay values (λ ≈ 0.006–0.01), indicating negligible heterogeneity. This aligns with the characteristics of PAM as dense motion sensor data without irregular sampling.
>
> | Variable | λ | Variable | λ |
> |------|-------------------------|------|-------------------------|
> | v0  | 1.0450868424778417 | v1  | 0.00886761257294687 |
> | v2  | 0.00661961076887879 | v3  | 0.052645291699899996 |
> | v4  | 0.009104271500002512 | v5  | 0.006600063023817579 |
> | v6  | 0.046407817780285375 | v7  | 0.04617853995740749 |
> | v8  | 0.008232738877070149 | v9  | 0.007195747311196391 |
> | v10 | 0.008363672881460414 | v11 | 0.027247531748140454 |
> | v12 | 0.01110076420421784  | v13 | 0.0072789112031877426 |
> | v14 | 0.008056690267908827 | v15 | 0.007563579241977809 |
> | v16 | 0.007136300359792959 | — | — |
>
> These results indicate that DBGL can generalize to diverse time series, even when the decay heterogeneity assumption does not hold.
>
>
> **Q4**: Computational analysis.
>
> **A4**: Thank you for your comment regarding computational cost. We provide both theoretical complexity analysis and empirical evaluation.
>
>
> **Time Complexity:**
> For an input of size O(B×T×V×D), where B=batch size, T=sequence length, V=number of variable nodes, and D=hidden dimension, the per-step operations are:
> - Codebook aggregation: $O((B+V)·D·N)  $
> - GNN message passing (EdgeSAGEConv + edge update): $O(L·E·(D+E))  $
> - Decay computation and gated update: $O(B·V·D) $
>
> Iterating over T steps gives total complexity:
> $O(T·((L·E·(D+E)) + B·V·D + (B+V)·D·N))$,
> where L=number of GNN layers, E=number of edges (≤ B·V for irregular time series), N=codebook size.
> All B·V·D operations are linear and efficiently parallelizable on GPUs.
>
> [1] Reiss, A., & Stricker, D. (2012, June). Introducing a new benchmarked dataset for activity monitoring. In 2012 16th international symposium on wearable computers (pp. 108-109). IEEE.

---

> ### Author Response · Authors · 2025-11-18
> **Response to Reviewer qRN1 (part 3/5)**
>
> **A4 Empirical Evaluation:**
> We measured GPU memory and training time per epoch on a single 24GB NVIDIA RTX 4090 using the P12 dataset. Baseline: B=256, V=36, L=215, C=4096. We varied one parameter at a time:
>
> We also conduct bottleneck analysis on a single 24GB NVIDIA GeForce RTX 4090 GPU using the P12 dataset, exploring the impact of batch size (B), codebook size (C), sequence length (L), and number of variables (V). To vary L and V, we repeated sequences and variables accordingly to increase sequence length and variable count. We report both GPU memory consumption and training time per epoch. Our baseline configuration is B=256, V=36, L=215, C=4096, and when varying one parameter, the others are kept constant.
>
> | B    | Space (MiB) | Time (s/epoch) | V    | Space (MiB) | Time (s/epoch) | L    | Space (MiB) | Time (s/epoch) | C    | Space (MiB) | Time (s/epoch) |
> |----|---|---|---|------|------|------|-------------|------|------|-----|-------|
> | 128  | 2670        | 56             | 36   | 3806        | 30             | 215  | 3806        | 30             | 512  | 2950        | 28             |
> | 256  | 3806        | 30             | 72   | 5258        | 35             | 430  | 6294        | 66             | 1024 | 3064        | 28             |
> | 512  | 6034        | 18             | 144  | 8154        | 48             | 860  | 11254       | 192            | 2048 | 3238        | 29             |
> | 1024 | 10776       | 12             | 288  | 14208       | 71             | 1290 | 16316       | 338            | 4096 | 3806        | 30             |
> | 2048 | 19700       | 9              | 432  | 20306       | 89             | 1720 | 21288       | 525            | 8172 | 4968        | 31             |
>
> **Summary:**
> - Sequence length L is the dominant factor affecting memory and training time.
> - Number of variables V also impacts both memory and time significantly.
> - Larger batch sizes B increase memory but reduce per-epoch time due to better GPU utilization.
> - Codebook size C has a negligible effect.
>
> Overall, despite additional overhead from codebook aggregation and EdgeSAGEConv, the method is computationally feasible and efficiently parallelizable for realistic clinical datasets.
>
> **Q5:** Related work summary.
>
> **A5:** Thank you for the suggestion. While methods such as SeFT [2], StraTS [3], and Warpformer [4] handle irregular sampling without resampling, they primarily operate on flattened sequences with local attention or interpolation. This representation captures limited patient–variable interactions (The individuality of the patient) and often overlooks the variable-specific temporal heterogeneity inherent in clinical data.
>
> Our DBGL framework addresses both limitations by:
> - Constructing patient–variable bipartite graphs to explicitly capture which variable is observed at each time point, preserving the structural irregularity.
> - Learning variable-specific decay rates, enabling adaptive modeling of heterogeneous temporal dynamics across variables.
>
> These design choices allow DBGL to represent clinical time series more faithfully, leading to improved downstream task performance compared to prior transformer-based approaches.
>
> We have revisited and revised the discussion of related work, and corrected previously misleading statements in our revised version, as detailed below:
>
> Irregular sampling is a central challenge in multivariate clinical time-series. Traditional interpolation or resampling pipelines produce regularly spaced sequences but may obscure informative missingness patterns. Recent transformer- and diffusion-based approaches incorporate continuous-time embeddings or learned interpolation to mitigate these issues, yet they still operate primarily on flattened sequences and therefore struggle to represent which variable is observed at which time, a key structural signal in clinical data. Bipartite variable–time graphs make this structure more explicit by linking variable nodes to time nodes, but they are typically constructed within local windows and do not capture patient-specific sampling behaviors. Moreover, clinical variables naturally exhibit heterogeneous temporal dynamics, some change within minutes, others over hours or days, yet most existing models encode time uniformly and lack explicit mechanisms for variable-specific decay.
>
> In summary, two main limitations remain in existing approaches. First, preprocessing or flattened sequence modeling can obscure inherent irregularity: resampling may eliminate informative missingness, and sequence-based variable modeling captures limited asynchronous interactions between patients and variables. Second, temporal dynamics are often treated uniformly across variables, ignoring that clinical variables evolve at heterogeneous rates and exhibit variable-specific sensitivity to elapsed time. This simplification can lead to incomplete or less informative patient representations for downstream clinical tasks.

---

> ### Author Response · Authors · 2025-11-18
> **Response to Reviewer qRN1 (part 4/5)**
>
> [2] Horn, M., Moor, M., Bock, C., Rieck, B., & Borgwardt, K. (2020, November). Set functions for time series. In International Conference on Machine Learning (pp. 4353-4363). PMLR.
>
> [3] Tipirneni, S., & Reddy, C. K. (2022). Self-supervised transformer for sparse and irregularly sampled multivariate clinical time-series. ACM Transactions on Knowledge Discovery from Data (TKDD), 16(6), 1-17.
>
> [4] Zhang, J., Zheng, S., Cao, W., Bian, J., & Li, J. (2023, August). Warpformer: A multi-scale modeling approach for irregular clinical time series. In Proceedings of the 29th ACM SIGKDD Conference on Knowledge Discovery and Data Mining (pp. 3273-3285).
>
> **Q6:** Computational Overhead and Runtime Complexity during inference stage.
>
> **A6:**Thank you for the question. During inference, patient–variable graphs are **dynamically constructed online** based on observed variables and timestamps, ensuring the graph reflects the actual irregular and incomplete measurements.
>
> To evaluate computational overhead, we compared DBGL with the sequence-based baseline KEDGN on a NVIDIA A800 80GB PCIe GPU. Training Time (s/epoch) and Inference times (seconds per epoch) are summarized below:
>
> | Training Time (s/epoch) | P19 | Physionet | MIMIC-III | P12 |
> |------------------------|-----|-----------|-----------|-----|
> | KEDGN                  | 6.6 | 8.4       | 36.6      | 17.4 |
> | DBGL                   | 10.8 | 9.6       | 50.4      | 30.0 |
>
> | Inference Time (s/epoch) | P19 | Physionet | MIMIC-III | P12 |
> |--------|------|-----------|-----------|------|
> | KEDGN | 1.76 | 0.93 | 5.92 | 1.61 |
> | DBGL  | 2.03 | 0.81 | 4.86 | 1.92 |
>
> The results show that dynamic graph generation introduces **minimal overhead**:
> - The graph at each step contains only the patient node and currently observed variable nodes.
> - Graph size scales linearly with the number of observed variables, i.e., **O(V_obs)** per timestep, and message passing is efficiently parallelized on GPU.
>
> Overall, across all datasets, the runtime difference between DBGL and sequence-based baselines is within 0.3–1.0 seconds, demonstrating that the proposed graph mechanism is both **practical and efficient** for real-world clinical scenarios. In some cases (Physionet, MIMIC-III), DBGL even runs faster than KEDGN.
>
> **Q7:** About codebook analysis.
>
> **A7:** Thank you for the question. The codebook in DBGL enables selective and similarity-aware cross-sample message passing. By softly projecting each variable embedding onto code vectors and aggregating messages from the nearest prototypes, it ensures that updates primarily incorporate information from samples with similar variable dynamics while suppressing dissimilar ones. This mechanism allows DBGL to capture cross-sample variable dependencies in a clinically coherent manner, independent of explicit labels.
>
> From Table 3, we observe that the matched code vectors bring modest gains on P19 and MIMIC-III but yield substantial improvements on more homogeneous datasets such as P12 and Physionet. This difference arises because in heterogeneous datasets (e.g., P19, MIMIC-III), variable dynamics are already diverse, and similarity-aware aggregation has a smaller relative impact. In contrast, in datasets with more uniform dynamics, the codebook significantly enhances message relevance and predictive performance.
>
> To further examine codebook usage, we compute the soft utilization rate, defined as the fraction of code vectors receiving above-uniform activation:
>
> Given the soft assignment weights $w\in \mathbb{R}^{N\times K}$, the utilization rate is defined as:
>
> $
> \bar{w}_k = \frac{1}{N} \sum_{i=1}^{N} w_{i,k}, \qquad
> \text{Utilization} = \frac{1}{K} \sum_{k=1}^{K} \mathbb{1} \left( \bar{w}_k > \frac{1}{K} \right).
> $
>
> This metric quantifies how many codebook entries are used more frequently than what a uniform distribution would predict.
> | Dataset | P19 | Physionet | MIMIC-III | P12 |
> |---------|------|-----------|-----------|------|
> | Utilization Rate | 0.4973 | 0.4977 | 0.4982 | 0.4978 |
>
> Key observations:
>
> No collapse occurs: utilization remains ~0.498 for all datasets, far from zero.
>
> Balanced engagement: code vectors are activated evenly across the codebook, showing stable and effective use.
>
> Consistent usage across datasets: despite differing dynamics, the codebook contributes meaningfully, particularly in homogeneous datasets where similarity-guided aggregation is more beneficial.
>
> These results confirm that the codebook is not merely a design embellishment but a robust mechanism that facilitates selective cross-sample communication, enhances learning of variable dependencies, and improves predictive performance when similarity structure is present.

---

> ### Author Response · Authors · 2025-11-18
> **Response to Reviewer qRN1 (part 5/5)**
>
> **Q8:** Comparison between Time Embedding versus Temporal Decay Encoding
>
> **A8:** Thank you for the question. In DBGL, Time Embedding (TE) encodes the observation timestamps and provides absolute or relative temporal context for each variable, forming the foundational input to the patient–variable graph. Time Decay Encoding (TDE), in contrast, models variable-specific decay rates, controlling how historical states influence the current representation. While both leverage temporal information, they capture distinct aspects: TE establishes the temporal framework, whereas TDE adjusts historical influence dynamically.
>
> To quantify their contributions, we conducted an ablation study comparing: the full model, the model without TE (W/O TE), and the model without TDE (W/O TDE). Results across four datasets are summarized below:
>
> | Model      | P19 AUROC    | P19 AUPRC    | Physionet AUROC | Physionet AUPRC | MIMIC-III AUROC | MIMIC-III AUPRC | P12 AUROC   | P12 AUPRC    |
> |------------|--------------|--------------|-----------------|-----------------|-----------------|-----------------|-------------|--------------|
> | W/O TE     | 86.8 ± 2.0   | 63.7 ± 1.7   | 88.2 ± 0.8      | 58.8 ± 2.4      | 85.0 ± 0.3      | 50.1 ± 0.3      | 87.1 ± 0.5  | 53.8 ± 1.9   |
> | W/O TDE    | 92.9 ± 0.7   | 64.3 ± 2.0   | 88.7 ± 0.8      | 59.1 ± 2.2      | 84.6 ± 1.1      | 48.2 ± 2.4      | 87.2 ± 0.5  | 53.3 ± 1.9   |
> | Full       | 93.3 ± 0.5   | 66.3 ± 1.4   | 89.1 ± 0.3      | 60.8 ± 2.2      | 85.2 ± 0.4      | 50.2 ± 1.0      | 88.1 ± 0.4  | 56.3 ± 1.0   |
>
>
> Key observations:
>
> TE effect: Removing TE causes a large drop in AUROC (e.g., P19 93.3 → 86.8), showing its critical role in encoding temporal context. Its effect on AUPRC is moderate but dataset-dependent.
>
> TDE effect: Removing TDE reduces AUPRC consistently across datasets (e.g., P12 56.3 → 53.3), indicating that dynamic weighting of historical states improves precision and recall for positive events. Its effect on AUROC is smaller than TE.
>
> Complementarity: TE provides the temporal framework, while TDE modulates historical influence based on variable-specific dynamics. Ablation results confirm that both are necessary: removing either leads to measurable performance degradation, highlighting their synergistic roles.
>
> In summary, TE and TDE capture complementary temporal aspects: TE encodes “when” observations occur, and TDE encodes “how past states decay.” Their combination enables DBGL to leverage both macro- and micro-level temporal information, resulting in robust and accurate predictions.

---

> ### Author Response · Authors · 2025-11-26
> **Supplement of repsonse to Reviewer qRN1**
>
> In TimeCHEAT [5], the time–variable bipartite graph updates each time node by aggregating all observed variables at that time, and each variable node by aggregating all time points where it is observed, so it performs better on the PAM dataset. While this structure captures variable–time interactions, it cannot explicitly model **variable-specific temporal decay**: clinical variables evolve on different time scales (e.g., heart rate changes every minute, creatinine over hours or days). Consequently, dependencies across consecutive time points are not dynamically encoded, limiting the model’s ability to represent fast- vs. slow-changing physiological signals.
>
> Modeling such temporal dependencies is critical in **clinical scenarios**, as it directly affects the model’s ability to capture the persistence and rate of change of patient states, which in turn influences the reliability of downstream clinical predictions.
>
> To address this, our proposed DBGL shifts the modeling perspective from variable–time to patient–variable, constructing a sequence of patient–variable bipartite graphs. At each time step:
>
> 1. Patient nodes connect to currently observed variable nodes and aggregate multi-variable information via graph message passing.
>
> 2. Over time, the patient node states are dynamically updated via decay functions and recurrent units, explicitly capturing temporal continuity and asynchronous variable changes.
>
> This design offers several advantages:
>
> 1. **Dynamic dependency modeling**: patient nodes share hidden states across consecutive time steps, enabling explicit modeling of temporal evolution rather than relying solely on sparse time-node aggregation.
>
> 2. **Variable-specific decay rate adaptation**: the decay mechanism combined with patient node memory allows fast-changing variables (e.g., vital signs) and slow-changing variables (e.g., lab measurements) to update at different rates within the unified graph, reflecting real physiological dynamics.
>
> 3. **Cross-variable information integration**: variable nodes are shared across patients, allowing the model to learn statistical dependencies at the population level, while patient nodes capture individual-specific patterns.
>
> We further compared DBGL with TimeCHEAT on classification performance (AUROC/AUPRC):
> | Method      | P19 AUROC | P19 AUPRC | P12 AUROC | P12 AUPRC |
> |------------|-----------|-----------|-----------|-----------|
> | TimeCHEAT  | 89.5 ± 1.9 | 56.1 ± 4.6 | 84.5 ± 0.7 | 48.2 ± 1.9 |
> | DBGL       | 93.3 ± 0.5 | 66.3 ± 1.4 | 88.1 ± 0.4 | 56.3 ± 1.0 |
> | Gains      | +3.8       | +10.2      | +3.6       | +8.1       |
>
> As shown, our approach achieves substantial performance gains, demonstrating that DBGL’s more sophisticated design effectively captures temporal continuity and asynchronous variation patterns, while leveraging cross-patient information for more accurate modeling of **irregular clinical time series**. This performance improvement is much greater than the gain achieved by TimeCheat on the PAM dataset.
>
> We hope that the proposed response can address your concerns and problems. We also hope to have further discussions with you and look forward to your reply.
>
> [5] Liu, J., Cao, M., & Chen, S. (2025, April). Timecheat: A channel harmony strategy for irregularly sampled multivariate time series analysis. In Proceedings of the AAAI Conference on Artificial Intelligence (Vol. 39, No. 18, pp. 18861-18869).

---

### Official Review · Reviewer_k3TC · 2025-10-29

**Soundness:** 2
**Presentation:** 2
**Contribution:** 2
**Rating:** 4
**Confidence:** 2

**Summary:**

This paper proposes DBGL, a novel decay-aware bipartite graph learning framework for irregular medical time series. DBGL models time series as patient-variable bipartite graphs to preserve irregular sampling patterns and introduces a node-specific temporal decay mechanism to handle variable-specific forgetting rates. Extensive experiments on four clinical datasets demonstrate that DBGL outperforms existing methods in prediction tasks and shows superior robustness under high missingness conditions.

**Strengths:**

1. Novel Bipartite Graph Formulation: It introduces an innovative patient-variable bipartite graph that directly encodes irregular sampling patterns without artificial alignment, effectively preserving crucial information about observation dependencies and missingness.

2. Adaptive, Variable-Specific Decay Mechanism: The model features a novel node-specific temporal decay encoding, allowing different clinical variables to "forget" information at unique, data-driven rates, which more accurately reflects real-world physiological processes compared to uniform decay assumptions.

3. Comprehensive Empirical Validation: DBGL demonstrates consistent and superior performance across four public clinical datasets, significantly outperforming a wide range of strong baselines and showing remarkable robustness in challenging "leave-variables-out" scenarios with high missingness.

**Weaknesses:**

1. Limited Scope of "Variable Decay" Evaluation: The paper's central claim is modeling "variable decay irregularity," where each clinical variable decays at a different rate. However, the evidence for this is primarily indirect, demonstrated through overall performance gains and ablation studies. A key missing analysis is a direct examination of the learned decay rates (λ_{p,n}^t). The work would be significantly strengthened by visualizing or statistically analyzing these rates across different variable types (e.g., vital signs vs. lab tests) to validate if they align with clinical intuition (e.g., heart rate changes minute-to-minute, while a creatinine level changes over hours or days). Without this, it remains unclear if the model is learning clinically meaningful, distinct decay patterns or simply using the mechanism as a flexible, yet uninterpretable, fitting tool.

2. Scalability and Complexity Concerns: The proposed method constructs a bipartite graph at each time step, which is a key to its success but also its primary computational burden. For long sequences with many patients and variables, this could lead to significant memory and time complexity, potentially limiting its application to real-time clinical settings or very large datasets. The paper would benefit from a more formal complexity analysis (e.g., O(|V| |E|) per time step) and a discussion of this trade-off. An actionable suggestion for future work would be to explore strategies for dynamic graph sparsification or sampling to improve scalability without a significant loss in performance.

3. Evaluation: A key weakness of the evaluation is its confinement to binary classification tasks, which fails to adequately demonstrate the generalizability and temporal modeling prowess of the proposed DBGL framework.

**Questions:**

See weaknesses.

---

> ### Author Response · Authors · 2025-11-18
> **Response to Reviewer k3TC (part 1/2)**
>
> **Q1**: Limited Scope of "Variable Decay" Evaluation
>
> **A1**: Thank you for raising this point. To directly assess whether our model captures clinically meaningful temporal dynamics, we conducted a systematic analysis of variable-specific decay rates (λ_{p,n}^t) across all datasets.
>
> **Decay Rate Definition:** Higher λ indicates fast-changing variables (e.g., HR, BP), while lower λ corresponds to more stable variables (e.g., creatinine, hemoglobin). This metric therefore provides a unified quantitative way to capture clinically meaningful differences in temporal behavior across variables.
>
> **Representative Results (P12 dataset):**
>
> | Variable | λ  | Variable | λ  | Variable | λ  |
> |----------|----|---------|----|---------|----|
> | HR       | 1.73 | Temp     | 0.0265 | Creatinine | 0.1399 |
> | Lactate  | 13.00 | ALP     | 14.71 | PaCO2   | 0.0225 |
> | ALT      | 14.62 | AST    | 13.89 | MAP     | 0.0913 |
>
> Kruskal–Wallis test confirms significant heterogeneity across variables (p = 3.87 × 10^{-118}), indicating that variables exhibit distinct temporal decay patterns. Fast-changing vital signs show high λ, while stable lab tests exhibit low λ, aligning with clinical intuition. Some lab tests (e.g., ALT, AST, lactate) can change rapidly under acute conditions, highlighting the necessity of variable-specific decay modeling.
>
> **Other datasets:** Similar patterns are observed in P19, Physionet, and MIMIC-III (all p ≪ 0.001). To maintain brevity in this rebuttal, we present only representative examples; the full analysis across all variables and datasets will be included in the revised manuscript.
>
> We also evaluated different continuous-time kernel functions (Gaussian, linear, exponential) for the decay modeling. Performance differences are minor, with MLP + exponential kernel slightly outperforming others, supporting the robustness of our approach.
>
> | Kernel        | P19 AUROC | P19 AUPRC | Physionet AUROC | Physionet AUPRC | MIMIC-III AUROC | MIMIC-III AUPRC | P12 AUROC | P12 AUPRC | Avg. AUROC | Avg. AUPRC |
> |---------------|-----------|-----------|------------------|------------------|------------------|------------------|------------|------------|-------------|-------------|
> | MLP + Linear   | 93.2 ± 0.7 | 65.6 ± 0.9 | 89.3 ± 0.4 | 60.8 ± 2.2 | 84.8 ± 0.2 | 49.5 ± 1.0 | 87.6 ± 0.3 | 54.2 ± 0.5 | 88.73 | 57.53 |
> | MLP + Gaussian | 93.3 ± 0.4 | 66.1 ± 1.6 | 89.3 ± 0.4 | 60.7 ± 2.4 | 84.6 ± 0.9 | 48.3 ± 2.5 | 87.7 ± 0.5 | 54.5 ± 1.3 | 88.73 | 57.40 |
> | MLP + Exp      | 93.3 ± 0.5 | 66.3 ± 1.4 | 89.1 ± 0.3 | 60.8 ± 2.2 | 85.2 ± 0.4 | 50.2 ± 1.0 | 88.1 ± 0.4 | 56.3 ± 1.0 | 88.93 | 58.40 |
> | Exp            | 92.9 ± 0.3 | 65.2 ± 0.6 | 88.8 ± 0.7 | 58.2 ± 1.9 | 84.9 ± 0.2 | 49.9 ± 0.5 | 87.0 ± 0.3 | 53.0 ± 1.0 | 88.40 | 56.58 |
>
> MLP here means to use the hidden features.
>
> **Conclusion:** These analyses validate that our model learns variable-specific decay patterns that are clinically meaningful and heterogeneous. Uniform decay assumptions would fail to capture these differences, whereas our variable-specific decay encoding module enables more accurate representation of clinical time series and improved downstream performance.
>
> **Q2**: Scalability and Complexity Concerns
>
> **A2**: Thank you for the suggestion. We provide a more formal complexity analysis and practical scalability evaluation.
>
> **Time Complexity:**
> For input size $O(B \times T \times V \times D)$, where B=batch size, T=time steps, V=variables, D=hidden dim:
>
> - Codebook aggregation: $O((B+V) \cdot D \cdot N)$
> - GNN message passing (EdgeSAGEConv + edge update): $O(L \cdot E \cdot (D+E))$
> - Decay computation and gated update: $O(B \cdot V \cdot D)$
>
> Total time complexity over T steps:
> $O\big(T \cdot ((L \cdot E \cdot (D+E)) + B \cdot V \cdot D + (B+V) \cdot D \cdot N)\big)$
> where L = number of GNN layers, E = number of edges (typically ≪ B·V in irregular clinical time series), and N = codebook size. Message passing and decay updates are linear in BVD and can be efficiently parallelized on GPUs.
>
> **Space Complexity:**
> Memory cost for inputs, hidden states, codebook, and edge attributes: $O(B \cdot T \cdot V \cdot D + E \cdot D + N \cdot D)$.

---

> ### Author Response · Authors · 2025-11-18
> **Response to Reviewer k3TC (part 2/2)**
>
> **A2: Empirical GPU Evaluation (24GB RTX 4090, P12 dataset):**
>
> | Param | Values | Memory (MiB) | Time (s/epoch) | Notes |
> |-------|--------|--------------|----------------|-------|
> | Batch size B | 128 → 2048 | 2670 → 19700 | 56 → 9 | Memory scales with B; time decreases due to GPU parallelism |
> | Variables V | 36 → 432 | 3806 → 20306 | 30 → 89 | Memory/time scale roughly linearly with V |
> | Sequence length L | 215 → 1720 | 3806 → 21288 | 30 → 525 | L is the most significant bottleneck |
> | Codebook C | 512 → 8172 | 2950 → 4968 | 28 → 31 | Minor effect |
>
> **Discussion:**
> - GPU memory is the main bottleneck; sequence length and number of variables have the largest impact.
> - Despite additional overhead from codebook aggregation and GNN, computations are parallelizable.
> - Future work could explore **dynamic graph sparsification, node/edge sampling** to improve scalability for real-time or larger-scale deployments without significant performance loss.
>
>
> **Q3**: Evaluation on more dataset.
>
> **A3**: To further evaluate the generalization of our proposed DBGL framework beyond binary clinical tasks, we conducted experiments on the PAM dataset [1], which contains 18 physical activities across 9 subjects, with 17 variables (IMU + heart rate) sampled over 600 time steps.
>
> We applied DBGL for classification. Unlike clinical datasets, PAM variables are dense and regularly sampled, with minimal variable-specific decay irregularity. Applying the decay-aware module here would introduce noise and degrade performance.
>
> | Model        | Accuracy (%)      | Precision (%)     | Recall (%)        | F1 score (%)     |
> |--------------|-----------------|-----------------|-----------------|----------------|
> | Transformer  | 83.5 ± 1.5       | 84.8 ± 1.5       | 86.0 ± 1.2       | 85.0 ± 1.3     |
> | Trans-mean   | 83.7 ± 2.3       | 84.9 ± 2.6       | 86.4 ± 2.1       | 86.4 ± 2.1     |
> | GRU-D        | 83.3 ± 1.6       | 84.6 ± 1.2       | 84.6 ± 1.2       | 84.8 ± 1.2     |
> | SeFT         | 67.1 ± 2.2       | 70.0 ± 2.4       | 68.2 ± 1.5       | 68.5 ± 1.8     |
> | mTAND        | 74.6 ± 4.3       | 74.3 ± 4.0       | 79.5 ± 2.8       | 76.8 ± 3.4     |
> | IP-Net       | 74.3 ± 3.8       | 75.6 ± 2.1       | 77.9 ± 2.2       | 76.6 ± 2.8     |
> | DGM²-O       | 82.4 ± 2.3       | 85.2 ± 1.2       | 83.9 ± 2.3       | 84.3 ± 1.8     |
> | MTGNN        | 83.4 ± 1.9       | 85.2 ± 1.7       | 86.1 ± 1.9       | 85.9 ± 2.4     |
> | RainDrop     | 88.5 ± 1.5       | 89.9 ± 1.5       | 89.9 ± 0.6       | 89.8 ± 1.0     |
> | ViTST        | 95.8 ± 1.3       | 96.2 ± 1.3       | 96.2 ± 1.3       | 96.5 ± 1.2     |
> | TimeCHEAT    | 96.5 ± 0.6       | 97.1 ± 0.5       | 96.9 ± 0.6       | 97.0 ± 0.5     |
> | DBGL         | 90.4 ± 0.9       | 92.5 ± 0.7       | 92.4 ± 0.9       | 92.4 ± 0.8     |
>
>
> DBGL achieves strong performance (92.4 F1), surpassing most irregular-time baselines. Its decay-aware module provides limited benefit here due to the lack of variable-wise decay heterogeneity, whereas ViTST [2] and TimeCHEAT [3] exploit strong generic sequence modeling (patching, global Transformers) suitable for dense continuous signals.
>
>
> Our decay-rate analysis confirms this: almost all variables exhibit extremely small decay values (λ ≈ 0.006–0.01), indicating negligible heterogeneity. This aligns with the characteristics of PAM as dense motion sensor data without irregular sampling.
>
> | Variable | λ | Variable | λ |
> |------|-------------------------|------|-------------------------|
> | v0  | 1.0450868424778417 | v1  | 0.00886761257294687 |
> | v2  | 0.00661961076887879 | v3  | 0.052645291699899996 |
> | v4  | 0.009104271500002512 | v5  | 0.006600063023817579 |
> | v6  | 0.046407817780285375 | v7  | 0.04617853995740749 |
> | v8  | 0.008232738877070149 | v9  | 0.007195747311196391 |
> | v10 | 0.008363672881460414 | v11 | 0.027247531748140454 |
> | v12 | 0.01110076420421784  | v13 | 0.0072789112031877426 |
> | v14 | 0.008056690267908827 | v15 | 0.007563579241977809 |
> | v16 | 0.007136300359792959 | — | — |
>
> These results indicate that DBGL can generalize to diverse time series, even when the decay heterogeneity assumption does not hold.
>
> [1] Reiss, A., & Stricker, D. (2012, June). Introducing a new benchmarked dataset for activity monitoring. In 2012 16th international symposium on wearable computers (pp. 108-109). IEEE.
>
> [2] Li, Z., Li, S., & Yan, X. (2023). Time series as images: Vision transformer for irregularly sampled time series. Advances in Neural Information Processing Systems, 36, 49187-49204.
>
> [3] Liu, J., Cao, M., & Chen, S. (2025, April). Timecheat: A channel harmony strategy for irregularly sampled multivariate time series analysis. In Proceedings of the AAAI Conference on Artificial Intelligence (Vol. 39, No. 18, pp. 18861-18869).

---

> ### Author Response · Authors · 2025-11-26
> **Supplement of response to Reviewer k3TC**
>
> In TimeCHEAT [3], the time–variable bipartite graph updates each time node by aggregating all observed variables at that time, and each variable node by aggregating all time points where it is observed, so it performs better on the PAM dataset. While this structure captures variable–time interactions, it cannot explicitly model **variable-specific temporal decay**: clinical variables evolve on different time scales (e.g., heart rate changes every minute, creatinine over hours or days). Consequently, dependencies across consecutive time points are not dynamically encoded, limiting the model’s ability to represent fast- vs. slow-changing physiological signals.
>
> Modeling such temporal dependencies is critical in **clinical scenarios**, as it directly affects the model’s ability to capture the persistence and rate of change of patient states, which in turn influences the reliability of downstream clinical predictions.
>
> To address this, our proposed DBGL shifts the modeling perspective from variable–time to patient–variable, constructing a sequence of patient–variable bipartite graphs. At each time step:
>
> 1. Patient nodes connect to currently observed variable nodes and aggregate multi-variable information via graph message passing.
>
> 2. Over time, the patient node states are dynamically updated via decay functions and recurrent units, explicitly capturing temporal continuity and asynchronous variable changes.
>
> This design offers several advantages:
>
> 1. **Dynamic dependency modeling**: patient nodes share hidden states across consecutive time steps, enabling explicit modeling of temporal evolution rather than relying solely on sparse time-node aggregation.
>
> 2. **Variable-specific decay rate adaptation**: the decay mechanism combined with patient node memory allows fast-changing variables (e.g., vital signs) and slow-changing variables (e.g., lab measurements) to update at different rates within the unified graph, reflecting real physiological dynamics.
>
> 3. **Cross-variable information integration**: variable nodes are shared across patients, allowing the model to learn statistical dependencies at the population level, while patient nodes capture individual-specific patterns.
>
> We further compared DBGL with TimeCHEAT on classification performance (AUROC/AUPRC):
> | Method      | P19 AUROC | P19 AUPRC | P12 AUROC | P12 AUPRC |
> |------------|-----------|-----------|-----------|-----------|
> | TimeCHEAT  | 89.5 ± 1.9 | 56.1 ± 4.6 | 84.5 ± 0.7 | 48.2 ± 1.9 |
> | DBGL       | 93.3 ± 0.5 | 66.3 ± 1.4 | 88.1 ± 0.4 | 56.3 ± 1.0 |
> | Gains      | +3.8       | +10.2      | +3.6       | +8.1       |
>
> As shown, our approach achieves substantial performance gains, demonstrating that DBGL’s more sophisticated design effectively captures temporal continuity and asynchronous variation patterns, while leveraging cross-patient information for more accurate modeling of **irregular clinical time series**. This performance improvement is much greater than the gain achieved by TimeCheat on the PAM dataset.
>
> We hope that the proposed response can address your concerns and problems. We also hope to have further discussions with you and look forward to your reply.

---

### Official Review · Reviewer_VcfD · 2025-10-30

**Soundness:** 3
**Presentation:** 3
**Contribution:** 3
**Rating:** 4
**Confidence:** 4

**Summary:**

This paper introduces an bipartite graph approach, establishing cross-patient and cross-series connections, for irregular medical time-series modeling.

**Strengths:**

- An interesting approach: a bipartite graph representation combined with irregular decay modeling of clinical variables
- Significant performance improvements (especially in AUPRC) in experiments

**Weaknesses:**

The graph representation seems to work well for existing patients, but how about new patients? I think comparing graph and non-graph methods in this setup is also critical to give a comprehensive evaluation of different modeling designs.

**Questions:**

- How could the proposed method handle unknown patients, which do not occur in the training data but has clinical signals in your test period? The graph-based approaches seem to in general outperform non-graph methods, I wonder whether this is related to the train-test division. Please explain clearly about your setup.
- How to align different patients whose signals may include in different time periods? Their clinical signals may have some connections but not recorded in the same global timestep. In practice, how you convert global timestamps recorded in the database to your time step $t$ that unifies multiple patients?
- How does using different groups of patients to construct the graph affect the modeling performance?
- How large the graph could your approach support given the current implementation? What is the potential bottleneck for scaling?

---

> ### Author Response · Authors · 2025-11-18
> **Response to Reviewer VcfD (part 1/2)**
>
> **Q1**: About New Patients rather than existing patients.
>
> **A1**: Thank you for your valuable comments. In fact, for the P12, P19, and Physionet datasets, we follow the prior work RAINDROP [1] and KEDGN [2] and perform patient-level data splitting. This means that each sample corresponds to one individual patient, and each patient appears only once in the dataset. Consequently, after the model is trained on the training set, all samples in the test set belong to previously unseen patients, directly addressing the scenario you raised.
>
> Under this setup, we compare graph-based and non-graph-based methods, which allows us to fairly and effectively evaluate the impact of different modeling designs. The results show that our graph-based model consistently outperforms non-graph methods on unseen patients.
>
> For the MIMIC-III dataset, the same patient may have multiple admissions, which is why this dataset is less strictly split at the patient level. Nevertheless, the primary evaluation on the first three datasets already demonstrates the effectiveness of our graph design on new patients.
>
> **Q2**: Deal with unknown patients and detials about the train-test division.
>
> **A2**: Thank you for your comment. In our DBGL framework, each patient is modeled as a patient–variable bipartite graph sequence. Edges indicate whether a variable has been observed. This design allows the model to accurately represent the observation patterns of unseen patients, even when their variable observations differ from all training samples, effectively handling irregular observations.
>
> For the P12, P19, and Physionet datasets, we follow prior works RAINDROP [1] and KEDGN [2] with patient-level splits: each patient appears exclusively in either the training, validation, or test set (split ratio 8:1:1). Therefore, all test samples correspond to previously unseen patients. For MIMIC-III, which is record-driven, we follow [2,3] and randomly split all samples into 70%/15%/15% for training, validation, and testing.
>
> The observed advantage of graph-based methods is not due to the train-test split. Rather, it stems from the ability of graph structures to naturally capture inter-variable relationships, aligning with clinical reasoning where multiple variables are jointly considered. Experiments show that our graph-based model consistently outperforms non-graph methods on new patients.
>
>
> **Q3**: Processing of irregular medical time series.
>
> **A3**: Thank you for your question. To harmonize clinical signals across patients with different admission times and sampling patterns, we follow a preprocessing protocol used in prior works [1-2], which combines local relative time alignment, patient-specific temporal grids, and sparse matrix representation:
>
> 1. **Local time alignment:** Each patient’s timestamps are converted into minutes since admission, creating a local time axis shared across all patients. We restrict all patients to a unified observation window (e.g., first 48 hours).
>
> 2. **Patient-specific temporal grids:** To preserve true sampling rhythms, we extract each patient’s unique observation times rather than forcing uniform intervals.
>
> 3. **Sparse matrix representation:** Each patient’s time series is stored as a 2D matrix (rows=time steps, columns=variables), with missing values indicated by a default value. True timestamps are preserved separately, enabling computation of Δt and capturing irregular dynamics.
>
> This preprocessing strategy effectively aligns heterogeneous and unaligned clinical signals while retaining patient-specific temporal irregularities, enabling unified modeling across multiple patients. Prior works and our experiments confirm its effectiveness.
>
> [1] Zhang, X., Zeman, M., Tsiligkaridis, T., & Zitnik, M. Graph-Guided Network for Irregularly Sampled Multivariate Time Series. In International Conference on Learning Representations.
>
> [2] Luo, Y., Liu, Z., Wang, L., Wu, B., Zheng, J., & Ma, Q. (2024). Knowledge-empowered dynamic graph network for irregularly sampled medical time series. Advances in Neural Information Processing Systems, 37, 67172-67199.
>
> [3] Silva, I., Moody, G., Scott, D. J., Celi, L. A., & Mark, R. G. (2012, September). Predicting in-hospital mortality of icu patients: The physionet/computing in cardiology challenge 2012. In 2012 computing in cardiology (pp. 245-248). IEEE

---

> ### Author Response · Authors · 2025-11-18
> **Response to Reviewer VcfD (part 2/2)**
>
> **Q4**: The effect of patient groups to construct the graph on performance.
>
> **A4**: Thank you for your comment. We evaluated how patient cohort composition affects model performance across four datasets: P12, P19, Physionet, and MIMIC-III. To simulate different patient distributions, we randomly split the training set into two non-overlapping subsets (50% each), trained separate models, and compared their performance:
>
> | AUPRC / AUROC | P19 | Physionet | MIMIC-III | P12 |
> | --- | --- | --- | --- | --- |
> | Subset 1 | 61.0±1.6 / 90.8±0.3 | 55.1±4.5 / 87.0±1.5 | 46.1±1.2 /83.7±0.5 | 50.8±1.0 / 85.9±0.6 |
> | Subset 2 | 61.6±1.2 / 91.0±0.2 | 54.6±3.3 /86.5±2.6 | 45.8±1.3 / 84.0±0.3 | 50.8±1.9 / 85.7±0.7 |
> | All | 66.3±1.4 / 93.3±0.5 | 60.8±2.2 / 89.1±0.3 | 50.2±1.0 / 85.2±0.4 | 56.3±1.0 / 88.1±0.4 |
>
> These results show that models trained on different subsets perform similarly, demonstrating robustness across patient distributions. Even with only 50% of training patients, our model outperforms nearly all baselines and is slightly below KEDGN.
>
> Since P12, P19, and Physionet include ICU type labels, we evaluated the model on different patient groups with specific ICU type to examine generalization:
>
> **P19 ICU Types**
> | ICU Type | Samples | AUROC |
> |----------|---------|-------|
> | Medical ICU | 3872 | 93.2±0.5 |
> | SIC Surgical ICU | 9 | 98.6±2.9 |
>
> **P12 ICU Types**
> | ICU Type | Samples | AUROC |
> |-----------|--------|-------|
> | Coronary Care Unit | 219 | 81.9±1.5 |
> | Cardiac Surgery Recovery Unit | 231 | 83.4±1.9 |
> | Medical ICU | 419 | 85.0±0.5 |
> | SIC Surgical ICU | 339 | 90.4±0.3 |
>
> **Physionet ICU Types**
> | ICU Type | Samples | AUROC |
> |-----------|--------|-------|
> | Coronary Care Unit | 64 | 90.3±2.4 |
> | Cardiac Surgery Recovery Unit | 91 | 99.4±0.6 |
> | Medical ICU | 153 | 84.8±0.5 |
> | SIC Surgical ICU | 91 | 85.4±1.2 |
>
> Across ICU types and datasets, our model maintains strong predictive performance. Extremely high AUROC values appear in small or highly homogeneous ICU groups (e.g., cardiac surgery recovery), which have consistent clinical patterns, while overall variance remains low. This demonstrates that our model generalizes well across heterogeneous patient cohorts and performs particularly strongly in clinically consistent populations.
>
> **Q5**: The potential bottleneck for scaling
>
> **A5**: The proposed patient–variable bipartite graph is constructed on a per-batch basis. Specifically, we conducted experiments on a single 24GB NVIDIA GeForce RTX 4090 GPU using the P12 dataset, exploring the impact of batch size (B), codebook size (C), sequence length (L), and number of variables (V). To vary L and V, we repeated sequences and variables accordingly to increase sequence length and variable count. We report both GPU memory consumption and training time per epoch. Our baseline configuration is B=256, V=36, L=215, C=4096, and when varying one parameter, the others are kept constant.
>
> | B    | Space (MiB) | Time (s/epoch) | V    | Space (MiB) | Time (s/epoch) | L    | Space (MiB) | Time (s/epoch) | C    | Space (MiB) | Time (s/epoch) |
> |------|-------------|----------------|------|-------------|----------------|------|-------------|----------------|------|-------------|----------------|
> | 128  | 2670        | 56             | 36   | 3806        | 30             | 215  | 3806        | 30             | 512  | 2950        | 28             |
> | 256  | 3806        | 30             | 72   | 5258        | 35             | 430  | 6294        | 66             | 1024 | 3064        | 28             |
> | 512  | 6034        | 18             | 144  | 8154        | 48             | 860  | 11254       | 192            | 2048 | 3238        | 29             |
> | 1024 | 10776       | 12             | 288  | 14208       | 71             | 1290 | 16316       | 338            | 4096 | 3806        | 30             |
> | 2048 | 19700       | 9              | 432  | 20306       | 89             | 1720 | 21288       | 525            | 8172 | 4968        | 31             |
>
> Increasing batch size B reduces training time but increases memory usage. Increasing number of variables V moderately increases both memory and time. Sequence length L has the most pronounced effect on memory and training time. In contrast, codebook size C has minimal impact (e.g., C from 512 → 8172 increases memory from 2950 → 4968 MiB, negligible change in time).
>
> **Scalability bottlenecks:** GPU memory is the primary constraint. Sequence length L and variable number V dominate memory usage, while large batch sizes can speed up training at the cost of memory. Codebook size C can be increased with minimal overhead.
>
> In practice, one can balance B, V, and L to fit available GPU resources. Our current implementation already supports graphs large enough for the datasets used in this work.
>
> These findings provide practical guidance for configuring model hyperparameters under limited GPU resources, balancing memory usage and training efficiency.

---

> > ### Comment · Reviewer_VcfD · 2025-11-25
> >
> > Thank you for your detailed feedback. Most of my concerns have been addressed. I have increased my rating for this work to 6.

---

> > > ### Author Response · Authors · 2025-11-25
> > > **Response to Reviewer VcfD**
> > >
> > > Thank you very much for your positive response.
> > >
> > > If you have any remaining questions or concerns regarding our responses or revisions, we would be very happy to further communicate and discuss them.
> > >
> > > We sincerely look forward to any additional feedback you may have, as it would be highly valuable for further improving our work.

---

### Author Response · Authors · 2025-11-18
**Revised version PDF submission.**

We sincerely thank all reviewers for their valuable comments and questions. We have addressed each concern point by point and revised the manuscript accordingly. **The revised version has been submitted, and all changes have been highlighted in blue for easier reference.** We greatly appreciate your time and effort, and we welcome any further questions or requests for clarification. Please feel free to raise them and engage in further discussion with us.

---

### Author Response · Authors · 2025-11-24

Dear ICLR 2026 AC, SAC, and PC,

We would like to express our gratitude to all the reviewers for their valuable feedback. We have carefully considered all suggestions and updated our submission accordingly.

However, we have not yet received responses from Reviewer VcfD, Reviewer k3TC, and Reviewer qRN1. At the same time, we are also very much looking forward to further discussions with the Reviewer nFwr. We believe that we have provided a thorough analysis and explanation of the existing opinions and concerns raised by all the reviewers.

We kindly request your assistance in reaching out to these reviewers. It would be greatly appreciated if you could encourage them to review our rebuttal, as we are eager to know if we have adequately addressed their questions and concerns.

We believe that constructive and timely communication between reviewers and authors is essential for the benefit of both parties.

Thank you for your hard work and support.

Best regards,

The authors of Paper 6101

---

### Author Response · Authors · 2025-11-27

Dear ICLR 2026 AC, SAC, and PC,

We would like to express our gratitude to all the reviewers for their valuable feedback. We have carefully considered all suggestions and updated our submission accordingly. As the rebuttal period is only one week left, we hope that there will still be some time for further discussion after we receive the reviewers' additional comments.

However, we have not yet received responses from Reviewer k3TC, and Reviewer qRN1. At the same time, we are also very much looking forward to further discussions with the Reviewer VcfD and Reviewer nFwr. We believe that we have provided a thorough analysis and explanation of the existing opinions and concerns raised by all the reviewers.

We kindly request your assistance in reaching out to these reviewers. It would be greatly appreciated if you could encourage them to review our rebuttal, as we are eager to know if we have adequately addressed their questions and concerns.

We believe that constructive and timely communication between reviewers and authors is essential for the benefit of both parties.

Thank you for your hard work and support.

Best regards,

The authors of Paper 6101

---

### Author Response · Authors · 2025-11-29

Dear ICLR 2026 AC, SAC, and PC,

Despite some unexpected circumstances during the rebuttal period, we would like to sincerely thank all reviewers for their insightful and constructive comments. We have carefully addressed each concern in our rebuttal and have updated the manuscript to improve clarity and completeness. All revisions are highlighted in blue for easy reference.

Importantly, **the core methodology of our work remains unchanged**. Based on the reviewers’ suggestions, we refined our explanations, strengthened the motivation with additional statistical analysis, and improved the presentation of our assumptions originally drawn from medical intuition. Furthermore, we included supplementary experiments—such as comparisons across patient groups, computational efficiency and bottleneck analysis, evaluation of different decay functions, and results on an additional dataset—to provide clearer evidence and a more comprehensive understanding of our approach. These additions serve to better illustrate the effectiveness and applicability of our method.

**We believe that these clarifications and supplemental analyses address the reviewers’ concerns and help present the contributions of our work more clearly and convincingly. We respectfully ask the committee to consider these updates when making the final decision.**

Thank you very much for your time and consideration.

---

### Meta-Review · Area_Chair_biNX · 2026-01-16

**Summary:**

The paper proposes a graph-based method for modeling patient-variable irregularities in clinical timeseries by using node-specific temporal decay encoding. The method is tested on 4 standard clinical benchmarks.


W1. A reviewer raised a question on the models' ability to handle new patients.

W2. Questions were raised concerning the effect of a different training cohort in graph construction (i.e. how robust are the resulting models).

W3. Scaling and computational complexity (construct a bipartite graph at each time step).

W4. Variable decay evaluation is limited in scope, and indirectly obtained via performance an ablation studies. Instead, the reviewer proposed direct visualization and statistical analysis of the learned decay rates for the different signals.

W5. Reviewer qRN1 contested the claim that transformer-based methods rely on resampling or imputation and pointed to several models that handle irregular sampling directly. Reviewer nFwr also pointed out that the methodological gaps the paper claims to address has been handled by other methods as well.

**Reviewer Concerns:**

W1. The authors pointed out that for most of the datasets, each sample is a different patient, thus, for those datasets, all patients in the test set are new to the model. This concern is addressed.

W2. The authors provided an additional experiment showing that the performance of the model is roughly the same for different training cohorts. Even though we do not know how similar the parameters are for the models trained on the different cohorts, I consider this concern addressed, as the goal seems to be attaining good performance rather than performing qualitative analysis of the resulting features.

W3. The authors point out that their graph building process is dynamic and efficient. They also provided a complexity analysis; while I can' 100% vouch for it, it looks reasonable on first sight. One of the reviewers ran the code and indicated that it is slow. I'll consider this concern addressed for small and medium size medical datasets, which should be acceptable for most medical applications. However, I ascertain that this technique could not be reasonably used for large scale models without significant changes. This translates to limitations in creating a potential foundation model using the proposed technique; as many clinical applications now rely on pre-trained foundation models adapted to a task, this is a non-negligible issue.

W4. The additional evaluation of decay for different signals is partially addressed, in that it is shown that the decay for the different features (HR,   lactate etc.) corresponds to clinical intuition. However, it is not shown how these decays adapt to the different patients; presumably, that is the entire point of the bipartite graph.

W5. Not addressed. The authors described the differences between their proposed method and existing techniques, pointing out the limitations of existing transformer-based methods for dealing with irregular data but no additional experiments were provided.


Other questions were raised concerning implementation details and interpretation of results. The authors were able to adequately resolve them.



The paper provides an interesting approach to modeling irregular time series, however, it does suffer from a lack of comparison to existing techniques, in particular recent transformer-based methods (see W5 above). Another drawback (albeit arguably acceptable) is the computational cost of applying the method, making it prohibitive for large scale dataset and constructing foundation models. Also, further qualitative analysis of the decay would be illuminating.

**Reviewer Scores:**

I have no way of knowing how the reviewers would have changed their scores.

---

### Decision · Program_Chairs · 2026-01-26

Reject